# Progressive Augmentation of GANs

**Dan Zhang**
Bosch Center for Artificial Intelligence
dan.zhang2@bosch.com

**Anna Khoreva**
Bosch Center for Artificial Intelligence
anna.khoreva@bosch.com

## Abstract

Training of Generative Adversarial Networks (GANs) is notoriously fragile, requiring to maintain a careful balance between the generator and the discriminator in order to perform well. To mitigate this issue we introduce a new regularization technique - *progressive augmentation of GANs (PA-GAN)*. The key idea is to gradually increase the task difficulty of the discriminator by progressively augmenting its input or feature space, thus enabling continuous learning of the generator. We show that the proposed progressive augmentation preserves the original GAN objective, does not compromise the discriminator's optimality and encourages a healthy competition between the generator and discriminator, leading to the better-performing generator. We experimentally demonstrate the effectiveness of PA-GAN across different architectures and on multiple benchmarks for the image synthesis task, on average achieving $\sim 3$ point improvement of the FID score.

## 1 Introduction

Generative Adversarial Networks (GANs) [11] are a recent development in the field of deep learning, that have attracted a lot of attention in the research community [27, 30, 2, 15]. The GAN framework can be formulated as a competing game between the generator and the discriminator. Since both the generator and the discriminator are typically parameterized as deep convolutional neural networks with millions of parameters, optimization is notoriously difficult in practice [2, 12, 24].

The difficulty lies in maintaining a healthy competition between the generator and discriminator. A commonly occurring problem arises when the discriminator overshoots, leading to escalated gradients and oscillatory GAN behaviour [23, 4]. Moreover, the supports of the data and model distributions typically lie on low dimensional manifolds and are often disjoint [1]. Consequently, there exists a nearly trivial discriminator that can perfectly distinguish real data samples from synthetic ones. Once such a discriminator is produced, its loss quickly converges to zero and the gradients used for updating parameters of the generator become useless. For improving the training stability of GANs regularization techniques [28, 12] can be used to constrain the learning of the discriminator. But as shown in [4, 18] they also impair the generator and lead to the performance degradation.

In this work we introduce a new regularization technique to alleviate this problem - *progressive augmentation of GANs (PA-GAN)* - that helps to control the behaviour of the discriminator and thus improve the overall training.[1] The key idea is to progressively augment the input of the discriminator network or its intermediate feature layers with auxiliary random bits in order to gradually increase the discrimination task difficulty (see Fig. 1). In doing so, the discriminator can be prevented from becoming over-confident, enabling continuous learning of the generator. As opposed to standard augmentation techniques (e.g. rotation, cropping, resizing), the proposed progressive augmentation does not directly modify the data samples or their features, but rather structurally appends to them. Moreover, it can also alter the input class. For instance, in the single-level augmentation the data sample or its features $x$ are combined with a random bit $s$ and both are provided to the discriminator.

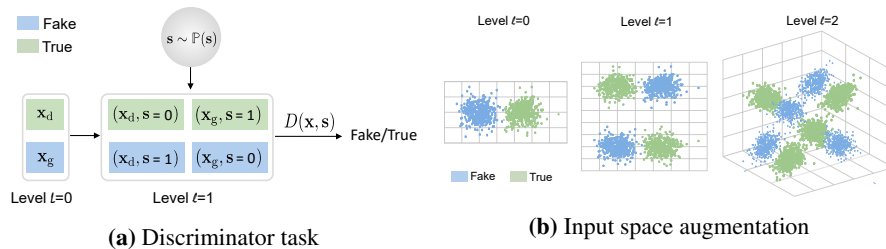

**(a)** Discriminator task

**(b)** Input space augmentation

**Figure 1:** Visualization of progressive augmentation. At level $l = 0$ (no augmentation) the discriminator $D$ aims at classifying the samples $\boldsymbol{x}_\mathrm{d}$ and $\boldsymbol{x}_\mathrm{g}$, respectively drawn from the data $\mathbb{P}_\mathrm{d}$ and generative model $\mathbb{P}_\mathrm{g}$ distributions, into true (green) and fake (blue). At single-level augmentation ($l = 1$) the class of the augmented sample is set based on the combination $\boldsymbol{x}_\mathrm{d}$ and $\boldsymbol{x}_\mathrm{g}$ with $s$, resulting in real and synthetic samples contained in both classes and leading to a harder task for $D$. With each extra augmentation level ($l \rightarrow l + 1$) the decision boundary between two classes becomes more complex and the discrimination task difficulty gradually increases. This prevents the discriminator from easily solving the task and thus leads to meaningful gradients for the generator updates.

The class of the augmented sample $(\boldsymbol{x}, s)$ is then set based on the combination $\boldsymbol{x}$ with $s$, resulting in real and synthetic samples contained in both classes, see Fig. 1-(a). This presents a more challenging task for the discriminator, as it needs to tell the real and synthetic samples apart plus additionally learn how to separate $(\boldsymbol{x}, s)$ back into $\boldsymbol{x}$ and $s$ and understand the association rule. We can further increase the task difficulty of the discriminator by progressively augmenting its input or feature space, gradually increasing the number of random bits during the course of training as depicted in Fig. 1-(b).

We prove that PA-GAN preserves the original GAN objective and, in contrast to prior work [1, 31, 30], does not bias the optimality of the discriminator (see Sec. 3.1). Aiming at minimum changes we further propose an integration of PA-GAN into existing GAN architectures (see Sec. 3.2) and experimentally showcase its benefits (see Sec. 4.1). Structurally augmenting the input or its features and mapping them to higher dimensions not only challenges the discrimination task, but, in addition, with each realization of the random bits alters the loss function landscape, potentially providing a different path for the generator to approach the data distribution.

Our technique is orthogonal to existing work, it can be successfully employed with other regularization strategies [28, 12, 30, 32, 6] and different network architectures [24, 35], which we demonstrate in Sec. 4.2. We experimentally show the effectiveness of PA-GAN for unsupervised image generation tasks on multiple benchmarks (Fashion-MNIST [34], CIFAR10 [17], CELEBA-HQ [15], and Tiny-ImageNet [7]), on average improving the FID score around 3 points. For PA combination with SS-GAN [6] we achieve the best FID of $14.7$ for the unsupervised setting on CIFAR10, which is on par with the results achieved by large scale BigGAN training [4] using label supervision.

## 2 Related Work

Many recent works have focused on improving the stability of GAN training and the overall visual quality of generated samples [28, 24, 35, 4]. The unstable behaviour of GANs is partly attributed to a dimensional mismatch or non-overlapping support between the real data and the generative model distributions [1], resulting in an almost trivial task for the discriminator. Once the performance of the discriminator is maxed out, it provides a non-informative signal to train the generator. To avoid vanishing gradients, the original GAN paper [11] proposed to modify the min-max based GAN objective to a non-saturating loss. However, even with such a re-formulation the generator updates tend to get worse over the course of training and optimization becomes massively unstable [1].

Prior approaches tried to mitigate this issue by using heuristics to weaken the discriminator, e.g. decreasing its learning rate, adding label noise or directly modifying the data samples. [30] proposed a one-sided label smoothing to smoothen the classification boundary of the discriminator, thereby preventing it from being overly confident, but at the same time biasing its optimality. [1, 31] tried to ensure a joint support of the data and model distributions to make the job of the discriminator harder by adding Gaussian noise to both generated and real samples. However, adding high-dimensional noise introduces significant variance in the parameter estimation, slowing down the training and requiring multiple samples for counteraction [28]. Similarly, [29] proposed to blur the input samples

and gradually remove the blurring effect during the course of training. These techniques perform direct modifications on the data samples.

Alternatively, several works focused on regularizing the discriminator. [12] proposed to add a soft penalty on the gradient norm which ensures a 1-Lipschitz discriminator. Similarly, [28] added a zero-centered penalty on the weighted gradient-norm of the discriminator, showing its equivalence to adding input noise. On the downside, regularizing the discriminator with the gradient penalty depends on the model distribution, which changes during training, and results in increased runtime due to additional gradient norm computation [18]. Most recently, [4] also experimentally showed that the gradient penalty may lead to the performance degradation, which corresponds to our observations as well (see Sec. 4.2) In addition to the gradient penalty, [4] also exploited the dropout regularization [32] on the final layer of the discriminator and reported its similar stabilizing effect. [24] proposed another way to stabilize the discriminator by normalizing its weights and limiting the spectral norm of each layer to constrain the Lipschitz constant. This normalization technique does not require intensive tuning of hyper-parameters and is computationally light. Moreover, [35] showed that spectral normalization is also beneficial for the generator, preventing the escalation of parameter magnitudes and avoiding unusual gradients.

Several methods have proposed to modify the GAN training methodology in order to further improve stability, e.g. by considering multiple discriminators [8], growing both the generator and discriminator networks progressively [15] or exploiting different learning rates for the discriminator and generator [13]. Another line of work resorts to objective function reformulation, e.g. by using the Pearson $\chi^2$ divergence [22], the Wasserstein distance [2], or f-divergence [25].

In this work we introduce a novel and orthogonal way of regularizing GANs by progressively increasing the discriminator task difficulty. In contrast to other techniques, our method does not bias the optimality of the discriminator or alter the training samples. Furthermore, the proposed augmentation is complementary to prior work. It can be employed with different GAN architectures and combined with other regularization techniques (see Sec. 4).

## 3 Progressive Augmentation of GANs

### 3.1 Theoretical Framework of PA-GAN

The core idea behind the GAN training [11] is to set up a competing game between two players, commonly termed discriminator and generator. The discriminator aims at distinguishing the samples $\boldsymbol{x} \in \mathcal{X}$ respectively drawn from the data distribution $\mathbb{P}_\mathrm{d}$ and generative model distribution $\mathbb{P}_\mathrm{g}$, i.e. performing binary classification $D : \mathcal{X} \mapsto [0, 1]$. [2] The aim of the generator, on the other hand, is to make synthetic samples into data samples, challenging the discriminator. In this work, $\mathcal{X}$ represents a compact metric space such as the image space $[-1, 1]^N$ of dimension $N$. Both $\mathbb{P}_\mathrm{d}$ and $\mathbb{P}_\mathrm{g}$ are defined on $\mathcal{X}$. The model distribution $\mathbb{P}_\mathrm{g}$ is induced by a function $G$ that maps a random vector $\boldsymbol{z} \sim \mathbb{P}_\mathrm{z}$ to a synthetic data sample, i.e. $\boldsymbol{x}_\mathrm{g} = G(\boldsymbol{z}) \in \mathcal{X}$. Mathematically, the two-player game is formulated as

$$\min_G \max_D \mathbb{E}_{\mathbb{P}_\mathrm{d}} \left\{ \log \left[ D(\boldsymbol{x}) \right] \right\} + \mathbb{E}_{\mathbb{P}_\mathrm{g}} \left\{ \log \left[ 1 - D(\boldsymbol{x}) \right] \right\}. \tag{1}$$

As being proved by [11], the inner maximum equals the Jensen-Shannon (JS) divergence between $\mathbb{P}_\mathrm{d}$ and $\mathbb{P}_\mathrm{g}$, i.e., $D_\mathrm{JS} \left( \mathbb{P}_\mathrm{d} \| \mathbb{P}_\mathrm{g} \right)$. Therefore, the GAN training attempts to minimize the JS divergence between the model and data distributions.

**Lemma 1.** *Let $s \in \{0, 1\}$ denote a random bit with uniform distribution $\mathbb{P}_\mathrm{s}(s) = \frac{\delta[s] + \delta[s-1]}{2}$, where $\delta[s]$ is the Kronecker delta. Associating $s$ with $\boldsymbol{x}$, two joint distributions of $(\boldsymbol{x}, s)$ are constructed as*

$$\mathbb{P}_\mathrm{x,s}(\boldsymbol{x}, s) \triangleq \frac{\mathbb{P}_\mathrm{d}(\boldsymbol{x}) \delta[s] + \mathbb{P}_\mathrm{g}(\boldsymbol{x}) \delta[s-1]}{2}, \quad \mathbb{Q}_\mathrm{x,s}(\boldsymbol{x}, s) \triangleq \frac{\mathbb{P}_\mathrm{g}(\boldsymbol{x}) \delta[s] + \mathbb{P}_\mathrm{d}(\boldsymbol{x}) \delta[s-1]}{2}. \tag{2}$$

*Their JS divergence is equal to*

$$D_\mathrm{JS} \left( \mathbb{P}_\mathrm{x,s} \| \mathbb{Q}_\mathrm{x,s} \right) = D_\mathrm{JS} \left( \mathbb{P}_\mathrm{d} \| \mathbb{P}_\mathrm{g} \right). \tag{3}$$

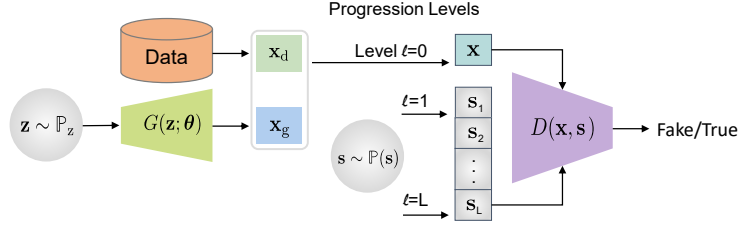

**Figure 2:** PA-GAN overview. With each level of progressive augmentation $l$ the dimensionality of $\boldsymbol{s}$ is enlarged from 1 to $L$, $\boldsymbol{s} = \{s_1, s_2, \ldots, s_L\}$. The task difficulty of the discriminator gradually increases as the length of $\boldsymbol{s}$ grows.

*Taking (2) as the starting point and with $\boldsymbol{s}_l$ being a sequence of i.i.d. random bits of length $l$, the recursion of constructing the paired joint distributions of $(\boldsymbol{x}, \boldsymbol{s}_l)$*

$$\mathbb{P}_{\mathrm{x},\mathbf{s}_l}(\boldsymbol{x}, \boldsymbol{s}_l) \triangleq \mathbb{P}_{\mathrm{x},\mathbf{s}_{l-1}}(\boldsymbol{x}, \boldsymbol{s}_{l-1})\delta[s_l]/2 + \mathbb{Q}_{\mathrm{x},\mathbf{s}_{l-1}}(\boldsymbol{x}, \boldsymbol{s}_{l-1})\delta[s_l - 1]/2$$
$$\mathbb{Q}_{\mathrm{x},\mathbf{s}_l}(\boldsymbol{x}, \boldsymbol{s}_l) \triangleq \mathbb{Q}_{\mathrm{x},\mathbf{s}_{l-1}}(\boldsymbol{x}, \boldsymbol{s}_{l-1})\delta[s_l]/2 + \mathbb{P}_{\mathrm{x},\mathbf{s}_{l-1}}(\boldsymbol{x}, \boldsymbol{s}_{l-1})\delta[s_l - 1]/2 \tag{4}$$

*results into a series of JS divergence equalities for $l = 1, 2, \ldots, L$, i.e.,*

$$D_{\mathrm{JS}}\left(\mathbb{P}_{\mathrm{d}} \| \mathbb{P}_{\mathrm{g}}\right) = D_{\mathrm{JS}}\left(\mathbb{P}_{\mathrm{x},\mathbf{s}_1} \| \mathbb{Q}_{\mathrm{x},\mathbf{s}_1}\right) = \cdots = D_{\mathrm{JS}}\left(\mathbb{P}_{\mathrm{x},\mathbf{s}_L} \| \mathbb{Q}_{\mathrm{x},\mathbf{s}_L}\right). \tag{5}$$

**Theorem 1.** *The min-max optimization problem of GANs [11] as given in (1) is equivalent to*

$$\min_G \max_D \mathbb{E}_{\mathbb{P}_{\mathrm{x},\mathbf{s}_l}} \left\{ \log\left[D(\boldsymbol{x}, \boldsymbol{s}_l)\right] \right\} + \mathbb{E}_{\mathbb{Q}_{\mathrm{x},\mathbf{s}_l}} \left\{ \log\left[1 - D(\boldsymbol{x}, \boldsymbol{s}_l)\right] \right\} \quad \forall l \in \{1, 2, \ldots, L\}, \tag{6}$$

*where the two joint distributions, i.e., $\mathbb{P}_{\mathrm{x},\mathbf{s}_l}$ and $\mathbb{Q}_{\mathrm{x},\mathbf{s}_l}$, are defined in (4) and the function $D$ maps $(\boldsymbol{x}, \boldsymbol{s}_l) \in \mathcal{X} \times \{0, 1\}^l$ onto $[0, 1]$. For a fixed $G$, the optimal $D$ is*

$$D^*(\boldsymbol{x}, \boldsymbol{s}_l) = \frac{\mathbb{P}_{\mathrm{x},\mathbf{s}_l}(\boldsymbol{x}, \boldsymbol{s}_l)}{\mathbb{P}_{\mathrm{x},\mathbf{s}_l}(\boldsymbol{x}, \boldsymbol{s}_l) + \mathbb{Q}_{\mathrm{x},\mathbf{s}_l}(\boldsymbol{x}, \boldsymbol{s}_l)} = \frac{\mathbb{P}_{\mathrm{d}}(\boldsymbol{x})}{\mathbb{P}_{\mathrm{d}}(\boldsymbol{x}) + \mathbb{Q}_{\mathrm{d}}(\boldsymbol{x})}, \tag{7}$$

*whereas the attained inner maximum equals $D_{\mathrm{JS}}\left(\mathbb{P}_{\mathrm{x},\mathbf{s}_l} \| \mathbb{Q}_{\mathrm{x},\mathbf{s}_l}\right) = D_{\mathrm{JS}}\left(\mathbb{P}_{\mathrm{d}} \| \mathbb{P}_{\mathrm{g}}\right)$ for $l = 1, 2, \ldots, L$.*

According to Theorem 1, solving (1) is interchangeable with solving (6). In fact, the former can be regarded as a corner case of the latter by taking $l = 0$ as the absence of the auxiliary bit vector $\boldsymbol{s}$. As the length $l$ of $\boldsymbol{s}$ increases, the input dimension of the discriminator grows accordingly. Furthermore, two classes to be classified consist of both the data and synthetic samples as illustrated in Fig. 1-(a). Note that, the mixture strategy of the distributions of two independent random variables in Lemma 1 can be extended for any generic random variables (see Sec. S.2.4 in the supp. material).

When solving (1), $G$ and $D$ are parameterized as deep neural networks and SGD (or its variants) is typically used for the optimization, updating their weights in an alternating or simultaneous manner, with no guarantees on global convergence. Theorem 1 provides a series of JS divergence estimation proxies by means of the auxiliary bit vector $\boldsymbol{s}$ that in practice can be exploited as a regularizer to improve the GAN training (see Sec. 4.1 for empirical evaluation). First, the number of possible combinations of the data samples with $\boldsymbol{s}_l$ grows exponentially with $l$, thus helping to prevent the discriminator from overfitting to the training set. Second, the task of the discriminator gradually becomes harder with the length $l$. The input dimensionality of $D$ becomes larger and as the label of $(\boldsymbol{x}, \boldsymbol{s}_{l-1})$ is altered based on the new random bit $s_l$ the decision boundary becomes more complicated (Fig. 1-b). Given that, progressively increasing $l$ can be exploited during training to balance the game between the discriminator and generator whenever the former becomes too strong. Third, when the GAN training performance saturates at the current augmentation level, adding one random bit changes the landscape of the loss function and may further boost the learning.

### 3.2 Implementation of PA-GAN

The min-max problem in (6) shares the same structure as the original one in (1), thus we can exploit the standard GAN training for PA-GAN, see Fig. 2. The necessary change only concerns the discriminator. It involves 1) using checksum principle as a new classification criterion, 2) incorporating $\boldsymbol{s}$ in addition to $\boldsymbol{x}$ as the network input and 3) enabling the progression of $\boldsymbol{s}$ during training.

**Checksum principle.** The conventional GAN discriminator assigns TRUE (0) / FAKE (1) class label based on $\boldsymbol{x}$ being either data or synthetic samples. In contrast, the discriminator $D$ in (6)

requires $s_l$ along with $x$ to make the decision about the class label. Starting from $l = 1$, the two class distributions in (2) imply the label-0 for $(x_\mathrm{d}, s = 0)$, $(x_\mathrm{g}, s = 1)$ and label-1 for $(x_\mathrm{d}, s = 1)$, $(x_\mathrm{g}, s = 0)$. The real samples are no longer always in the TRUE class, and the synthetic samples are no longer always in the FAKE class, see Fig. 1-(a). To detect the correct class we can use a simple checksum principle. Namely, let the data and synthetic samples respectively encode bit $0$ and $1$ followed by associating the checksum $0(1)$ of the pair $(x, s)$ with TRUE(FAKE). [3] For more than one bit, $\mathbb{P}_{\mathrm{x},\mathbf{s}_l}$ and $\mathbb{Q}_{\mathrm{x},\mathbf{s}_l}$ are recursively constructed according to (4). Based on the checksum principle for the single bit case, we can recursively show its consistency for any bit sequence length $s_l, l > 1$. This is a desirable property for progression. With the identified checksum principle, we further discuss a way to integrate a sequence of random bits $s_l$ into the discriminator network in a progressive manner.

**Progressive augmentation.** With the aim of maximally reusing existing GAN architectures we propose two augmentation options. The first one is *input space augmentation*, where $s$ is directly concatenated with the sample $x$ and both are fed as input to the discriminator network. The second option is *feature space augmentation*, where $s$ is concatenated with the learned feature representations of $x$ attained at intermediate hidden layers. For both cases, the way to concatenate $s$ with $x$ or its feature maps is identical. Each entry $s_l$ creates one augmentation channel, which is replicated to match the spatial dimension of $x$ or its feature maps. Depending on the augmentation space, either the input layer or the hidden layer that further processes the feature maps will additionally take care of the augmentation channels along with the original input. In both cases, the original layer configuration (kernel size, stride and padding type) remains the same except for its channel size being increased by $l$. All the other layers of the discriminator remain unchanged. When a new augmentation level is reached, one extra input channel of the filter is instantiated to process the bit $l + 1$.

These two ways of augmentation are beneficial as they make the checksum computation more challenging for the discriminator, i.e., making the discriminator unaware about the need of separating $x$ and $s$ from the concatenated input. We note that in order to take full advantage of the regularization effect of progressive augmentation, $s$ needs to be involved in the decision making process of the discriminator either through input or feature space augmentation. Augmenting $s$ with the output $D(x)$ makes the task trivial, thereby disabling the regularization effect of the progressive augmentation. In this work we only exploit $s$ by concatenating it with either the input or the hidden layers of the network. However, it is also possible to combine it with other image augmentation strategies, e.g. using $s$ as an indicator for the rotation angle, as in [6], or the type of color augmentation that is imposed on the input $x$ and encouraging $D$ to learn the type through the checksum principle.

**Progression scheduling.** To schedule the progression we rely on the kernel inception distance (KID) introduced by [3] to decide if the performance of $G$ at the current augmentation level saturates or even starts degrading (typically happens when $D$ starts overfitting or becomes too powerful). Specifically, after $t$ discriminator iterations, we evaluate KID between synthetic samples and data samples drawn from the training set. If the current KID score is less than $5\%$ of the average of the two previous evaluations attained at the same augmentation level, the augmentation is leveled up, i.e. $l \to l + 1$. To validate the effectiveness of this scheduling mechanism we exploit it for the learning rate adaptation as in [3] and compare it with progressive augmentation in the next section.

## 4 Experiments

*Datasets:* We consider four datasets: Fashion-MNIST [34], CIFAR10 [17], CELEBA-HQ ($128 \times 128$) [15] and Tiny-ImageNet (a simplified version of ImageNet [7]), with the training set sizes equal to 60k, 50k, 27k and 100k plus the test set sizes equal to 10k, 10k, 3k, and 10k, respectively. Note that we focus on unsupervised image generation and do not use class label information.

*Networks:* We employ SN DCGAN [24] and SA GAN [35], both using spectral normalization (SN) [24] in the discriminator for regularization. SA GAN exploits the ResNet architecture with a self-attention (SA) layer [35]. Its generator additionally adopts self-modulation BN (sBN) [5] together with SN. We exploit the implementations provided by [18, 35]. Following [24, 35], we train SN DCGAN and SA GAN [35] with the non-saturation (NS) and hinge loss, respectively.

*Evaluation metrics:* We use Fréchet inception distance (FID) [14] as the main evaluation metric. Additionally, we also report inception score (IS) [33] and kernel inception distance (KID) [3] in

**Table 1:** FID improvement of `PA` across different datasets and network architectures. We experiment with augmenting the input and feature spaces, see Sec.4.1 for details.

| Method | PA | F-MNIST | CIFAR10 | CELEBA-HQ | T-ImageNet | $\Delta$PA |
|---|---|---|---|---|---|---|
| SN DCGAN [24] | ✗ | 10.6 | 26.0 | 24.3 | - | |
| | input | **6.2** | **22.2** | 20.8 | - | 4.2 |
| | feat | **6.2** | 22.6 | **18.8** | - | |
| SA GAN (sBN) [35] | ✗ | - | 18.8 | 17.8 | 47.6 | |
| | input | - | **16.1** | **15.4** | 44.8 | 2.6 |
| | feat | - | 16.3 | 15.8 | **44.7** | |

Sec. S7 of the supp. material. All measures are computed based on the same number of the test data samples and synthetic samples, following the evaluation framework of [21, 18]. By default all reported numbers correspond to the median of five independent runs with 300k, 500k, 400k and 500k training iterations for Fashion-MNIST, CIFAR10, CELEBA-HQ, and Tiny-ImageNet, respectively.

***Training details:*** We use uniformly distributed noise vector $z \in [-1, 1]^{128}$, the mini-batch size of $64$, and Adam optimizer [16]. The two time-scale update rule (TTUR) [13] is considered when choosing the learning rates for $D$ and $G$. For progression scheduling KID[4] is evaluated using samples from the training set every $t = 10$k iterations, except for Tiny-ImageNet with $t = 20$k given its approximately $2\times$ larger training set. More details are provided in Sec. S8 of the supp. material.

### 4.1 PA Across Different Architectures and Datasets

Table 1 gives an overview of the FID performance achieved with and without applying the proposed progressive augmentation (PA) across different datasets and networks. We observe consistent improvement of the FID score achieved by `PA` with both the input `PA` (`input`) and feature `PA` (`feat`) space augmentation (see the supp. material for augmentation details and ablation study on the augmentation space). From `SN DCGAN` to the ResNet-based `SA GAN` the FID reduction preserves approximately around 3 points, showing that the gain achieved by `PA` is complementary to the improvement on the architecture side. In comparison to input space augmentation, augmenting intermediate level features does not overly simplify the discriminator task, paralysing `PA`. In the case of `SN DCGAN` on CELEBA-HQ, it actually outperforms the input space augmentation. Overall, a stable performance gain of `PA`, independent of the augmentation space choice, showcases high generalization quality of `PA` and its easy adaptation into different network designs.[5]

Lower FID values achieved by `PA` can be attributed mostly to the improved sample diversity. By looking at generated images in Fig. 3 (and Fig. S4 in the supp. material), we observe that `PA` increases the variation of samples while maintaining the same image fidelity. This is expected as `PA` being a regularizer does not modify the GAN architecture, as in PG-GAN [15] or BigGAN [4], to directly improve the visual quality. Specifically, Fig. 3 shows synthetic images produced by `SN DCGAN` and `SA GAN` with and without PA, on Fashion-MNIST and CELEBA-HQ. By polar interpolation between two samples $z_1$ and $z_2$, from left to right we observe the clothes/gender change. `PA` improves sample variation, maintaining representative clothes/gender attributes and achieving smooth transition between samples (e.g. hair styles and facial expressions). For further evaluation, we also measure the diversity of generated samples with the MS-SSIM score [26]. We use 10k synthetic images generated with `SA GAN` on CELEBA-HQ. Employing `PA` reduces MS-SSIM from $0.283$ to $0.266$, while PG-GAN [15] achieves $0.283$, and MS-SSIM of 10k real samples is $0.263$.

**Comparison with SotA on Human Face Synthesis.** Deviating from from low- to high-resolution human face synthesis, the recent work COCO-GAN [19] outperformed PG-GAN [15] on the CELEBA dataset [20] via conditional coordinating. At the resolution $64$ of CELEBA, `PA` improves the `SA GAN` FID from $4.11$ to $3.35$, being better than COCO-GAN, which achieves FID of $4.0$ and outperforms PG-GAN at the resolution $128$ (FID of $5.74$ vs. $7.30$). Thus we conclude that the quality of samples generated by `PA` is comparable to the quality of samples generated by the recent state-of-the-art models [19, 15] on human face synthesis.

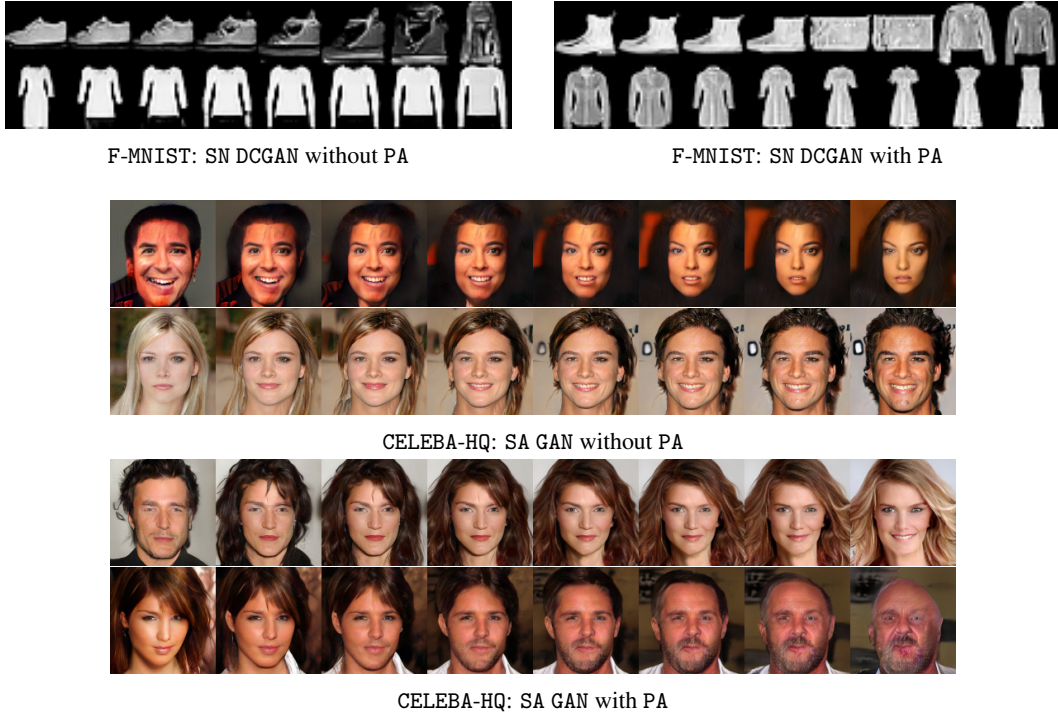

F-MNIST: SN DCGAN without PA              F-MNIST: SN DCGAN with PA

CELEBA-HQ: SA GAN without PA

CELEBA-HQ: SA GAN with PA

**Figure 3:** Synthetic images generated through latent space interpolation with and without using PA. PA helps to improve variation across interpolated samples, i.e., no close-by images looks alike.

**Ablation Study.** In Fig. 4 and Table 2 we present an ablation study on PA, comparing single-level augmentation (without progression) with progressive multi-level PA, showing the benefit of progression. From no augmentation to the first level augmentation, the required number of iterations varies over the datasets and architectures (30k∼ 70k). Generally the number of reached augmentation levels is less than 15. Fig. 4 also shows that single-level augmentation already improves the performance over the baseline SN DCGAN. However, the standard deviation of its FIDs across five independent runs starts increasing at later iterations. By means of progression, we can counteract this instability, while reaching a better FID result. Table 2 further compares augmentation at different levels with and without continuing with progression. Both augmentation and progression are beneficial, while progression alleviates the need of case dependent tuning of the augmentation level.

As a generic mechanism to monitor the GAN training, progression scheduling is usable not only for augmentation level-up, but also for other hyperparameter adaptations over iterations. Analogous to [3] here we test it for the learning rate adaptation. From Fig. 4, progression scheduling shows its effectiveness in assisting both the learning rate adaptation and PA for an improved FID performance. PA outperforms learning rate adaptation, i.e. median FID 22.2 vs. 24.0 across five independent runs.

**Regularization Effect of PA.** Fig. 5 depicts the discriminator loss ($D$ loss) and the generator loss ($G$ loss) behaviour as well as the FID curves over iterations. It shows that the discriminator of SN DCGAN very quickly becomes over-confident, providing a non-informative backpropagation signal to train the generator and thus leading to the increase of the $G$ loss. PA has a long lasting regularization effect on SN DCGAN by means of progression and helps to maintain a healthy competition between its discriminator and generator. Each rise of the $D$ loss and drop of the $G$ loss coincides with an iteration at which the augmentation level increases, and then gradually reduces after the discriminator timely adapts to the new bit. Observing the behaviour of the $D$ and $G$ losses, we conclude that both PA (input) and PA (feat) can effectively prevent the SN DCGAN discriminator from overfitting, alleviating the vanishing gradient issue and thus enabling continuous learning of the generator. At the level one augmentation, both PA (feat) and PA (input) start from the similar overfitting stage, i.e., ($a$) and ($b$) respectively at the iteration 60k and 70k. Combining the bit $s$ directly with high-level features eases the checksum computation. As a result, the $D$ loss of PA ($feat_{N/8}$) reduces faster, but making its future task more difficult due to overfitting to the previous augmentation level. On the other hand, PA (input) let the bits pass through all layers, and thus its adaptation to augmentation

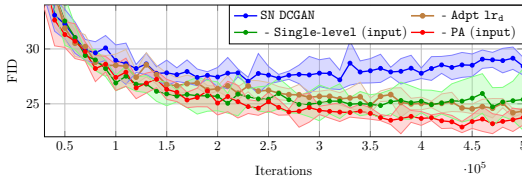

**Figure 4:** FID learning curves on SN DCGAN CIFAR10. The curves show the mean FID with one standard deviation across five random runs.

**Table 2:** Median FIDs of input space augmentation starting from the level $l$ with and without progression on CIFAR10 with SN DCGAN.

| Augment. level $l$ | Progression ✗ | Progression ✓ | $\Delta$PA |
|---|---|---|---|
| 0 | 26.0 | **22.2** | 3.8 |
| 1 | 23.8 | 22.3 | 1.5 |
| 2 | 23.6 | 22.9 | 0.7 |
| 3 | 23.5 | 22.9 | 0.6 |
| 4 | 23.5 | 23.2 | 0.3 |

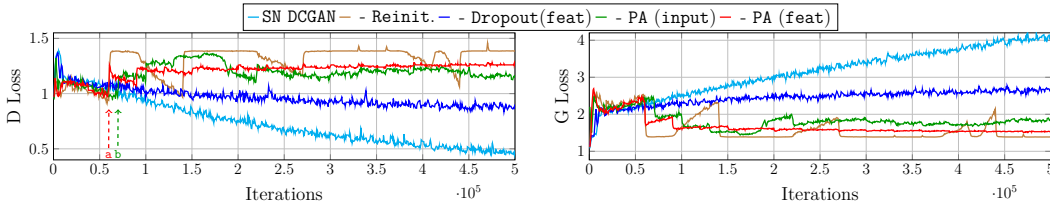

**(a)** Discriminator ($D$) and generator ($G$) loss over iterations

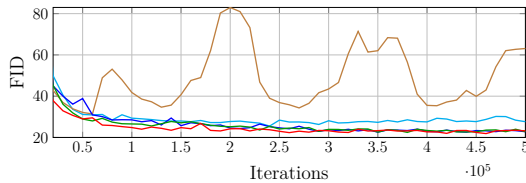

**(b)** FID over iterations

**Figure 5:** Behaviour of the discriminator loss ($D$ loss) and the generator loss ($G$ loss) as well as FID changes over iterations, using SN DCGAN on CIFAR10. PA acts as a stochastic regularizer, preventing the discriminator from becoming overconfident.

progression improves over iterations. In the end, both PA (feat) and PA (input) lead to similar regularization effect and result in the improved FID scores.

In Fig. 5 we also evaluate the Dropout [32] regularization applied on the fourth convolutional layer with the keep rate 0.7 (the best performing setting in our experiments). Both Dropout and PA resort to random variables for regularization. The former randomly removes features, while the latter augments them with additional random bits and adjusts accordingly the class label. In contrast to Dropout, PA has a stronger regularization effect and leads to faster convergence (more rapid reduction of FID scores). In addition, we compare PA with the Reinit. baseline, where at each scheduled progression all weights are reinitialized with Xavier initialization [10]. Compared to PA, using Reinit. strategy leads to longer adaptation time (the $D$ loss decay is much slower) and oscillatory GAN behaviour, thus resulting in dramatic fluctuations of FID scores over iterations.

## 4.2 Comparison and Combination with Other Regularizers

We further compare and combine PA with other regularization techniques, i.e., one-sided label smoothing [30], GP from [12], its zero-centered alternative GP_zero-cent from [28], Dropout [32], and self-supervised GAN training via auxiliary rotation loss (SS) [6].

One-sided label smoothing (Label smooth.) weakens the discriminator by smoothing its decision boundary, i.e., changing the positive labels from one to a smaller value. This is analogous to introducing label noise for the data samples, whereas PA alters the target labels based on the deterministic checksum principle. Benefiting from a smoothed decision boundary, Label smooth. slightly improves the performance of SN DCGAN (26.0 vs. 25.8), but underperforms in comparison to PA (input) (22.2) and PA (feat) (22.6). By applying PA on top of Label smooth. we observe a similar reduction of the FID score (23.1 and 22.3 for input and feature space augmentation, respectively).

Both GP and GP_zero-cent regularize the norms of gradients to stabilize the GAN training. The former aims at a 1-Lipschitz discriminator, and the latter is a closed-form approximation of adding input noise. Table 3 shows that both of them are compatible with PA but degrade the performance of SN DCGAN alone and its combination with PA. This effect has been also observed in [18, 4], constraining the

**Table 3:** FID performance of `PA`, different regularization techniques and their combinations on CIFAR10, see Sec. 4.2 for details.

| Method | PA | GAN | -Label smooth. [30] | -GP [12] | -GP$_{zero-cent}$ [28] | -Dropout [32] | -SS [6] | $\overline{\Delta PA}$ |
|---|---|---|---|---|---|---|---|---|
| SN DCGAN [24] | ✗ | 26.0 | 25.8 | 26.7 | 26.5 | 22.1 | — | |
| | input | 22.2 | 23.1 | 21.8 | 22.3 | 21.9 | — | 3.0 |
| | feat | 22.6 | 22.3 | 22.7 | 23.0 | **20.6** | — | 3.1 |
| SA GAN (sBN) [35] | ✗ | 18.8 | — | 17.8 | 17.8 | 16.2 | 15.7 | |
| | input | 16.1 | — | 15.8 | 16.1 | 15.5 | **14.7** | 1.3 |
| | feat | 16.3 | — | 16.1 | 15.9 | 15.6 | 14.9 | 1.3 |
| | $\overline{\Delta PA}$ | 3.1 | 3.1 | 3.2 | 2.8 | 0.8 | 0.9 | 2.3 |

learning of the discriminator improves the GAN training stability but at the cost of performance degradation. Note that, however, with `PA` performance degradation is smaller.

`Dropout` shares a common stochastic nature with `PA` as illustrated in Fig. 5 and in the supp. material. We observe from Table 3 that `Dropout` and `PA` can be both exploited as effective regularizers. `Dropout` acts locally on the layer. The layer outputs are randomly and independently subsampled, thinning the network. In contrast, `PA` augments the input or the layer with extra channels containing random bits, these bits also change the class label of the input and thus alter the network decision process. `Dropout` helps to break-up situations where the layer co-adapts to correct errors from prior layers and enables the network to timely re-learn features of constantly changing synthetic samples. `PA` regularizes the decision process of $D$, forcing $D$ to comprehend the input together with the random bits for correct classification and has stronger regularization effect than `Dropout`, see Fig. 5 and the supp. material. Hence, they have different roles. Their combination further improves FID by $\sim 0.8$ point on average, showing the complementarity of both approaches. It is worth noting that `Dropout` is sensitive to the selection of the layer at which it is applied. In our experiments (see the supp. material) it performs best when applied at the fourth convolutional layer.

Self-supervised training (SS-GAN) in [6] regularizes the discriminator by encouraging it to solve an auxiliary image rotation prediction task. From the perspective of self-supervision, `PA` presents the discriminator a checksum computation task, whereas telling apart the data and synthetic samples becomes a sub-task. Rotation prediction task was initially proposed and found useful in [9] to improve feature learning of convolutional networks. The checksum principle is derived from Theorem 1. Their combination is beneficial and achieves the best FID of $14.7$ for the unsupervised setting on CIFAR10, which is the same score as in the supervised case with large scale BigGAN training [4].

Overall, we observe that `PA` is consistently beneficial when combining with other regularization techniques, independent of input or feature space augmentation. Additional improvement of the FID score can come along with fine selection of the augmentation space type.

## 5  Conclusion

In this work we have proposed progressive augmentation (PA) - a novel regularization method for GANs. Different to standard data augmentation our approach does not modify the training samples, instead it progressively augments them or their feature maps with auxiliary random bits and casts the discrimination task into the checksum computation. PA helps to entangle the discriminator and thus to avoid its early performance saturation. We experimentally have shown consistent performance improvements of employing PA-GAN across multiple benchmarks and demonstrated that PA generalizes well across different network architectures and is complementary to other regularization techniques. Apart from generative modelling, as a future work we are interested in exploiting PA for semi-supervised learning, generative latent modelling and transfer learning.

## Footnotes

[1]https://github.com/boschresearch/PA-GAN

[2] $D(x)$ aims to learn the probability of $\boldsymbol{x}$ being true or fake, however, it can also be regarded as the sigmoid response of classification with cross entropy loss.

[3] By checksum we mean the XOR operation over a bit sequence.

[4]FID is used as the primary metric, KID is chosen for scheduling to avoid over-optimizing towards FID.

[5]We also experiment with using PA for WGAN-GP [2], improving FID from $25.0$ to $23.9$ on CIFAR10, see Sec. S.3.4 in the supp. material.

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
