[Supplementary Material]

# Supplementary Material
# Progressive Augmentation of GANs

**Dan Zhang**
Bosch Center for Artificial Intelligence
dan.zhang2@bosch.com

**Anna Khoreva**
Bosch Center for Artificial Intelligence
anna.khoreva@bosch.com

## S1   Content

This document completes the presentation of PA-GAN in the main paper with the following:

- Theoretical proofs for Lemma 1 and Theorem 1 in Sec. S2;
- Implementation details of PA-GAN in Sec. S3;
- Additional ablation studies in Sec. S4;
- Analysis of PA effectiveness as regularizer on the toy example in Sec. S5;
- Exemplar synthetic images in Sec. S6;
- Results for the IS [20] and KID [2] metrics in Sec. S7;
- Network architectures and hyperparameter settings in Sec. S8.

## S2   Theoretical Framework of PA-GAN

### S2.1   Information Theory Viewpoint on the JS Divergence

Apart from quantifying distributions' similarity, the JS divergence has an information theory interpretation that inspires our approach. In accordance with the binary classification task of the discriminator, we introduce a binary random variable $s$ with a uniform distribution $\mathbb{P}_s$. Associating $s = 0$ and $s = 1$ respectively with $\boldsymbol{x} \sim \mathbb{P}_d$ and $\boldsymbol{x} \sim \mathbb{P}_g$, we obtain a joint distribution function

$$\mathbb{P}_{x,s}(\boldsymbol{x}, s) \triangleq \frac{\mathbb{P}_d(\boldsymbol{x})\delta[s] + \mathbb{P}_g(\boldsymbol{x})\delta[s-1]}{2}, \tag{1}$$

where $\delta[\cdot]$ stands for the Kronecker delta function. The marginal distribution of $\mathbb{P}_{x,s}$ with respect to $\boldsymbol{x}$ (a.k.a. the mixture distribution) is equal to

$$\mathbb{P}_m \triangleq \mathbb{P}_s(s = 0)\mathbb{P}_d + \mathbb{P}_s(s = 1)\mathbb{P}_g = \frac{\mathbb{P}_d + \mathbb{P}_g}{2}. \tag{2}$$

Computing the mutual information of the two random variables $s$ and $\boldsymbol{x}$ based on $\mathbb{P}_{x,s}$ is identical to computing the JS divergence between $\mathbb{P}_d$ and $\mathbb{P}_g$, i.e.,

$$I(\boldsymbol{x}; s) = \frac{E_{\mathbb{P}_m}\left[p_d(\boldsymbol{x})\log p_d(\boldsymbol{x})\right] + E_{\mathbb{P}_m}\left[p_g(\boldsymbol{x})\log p_g(\boldsymbol{x})\right]}{2} = D_{JS}\left(\mathbb{P}_d\|\mathbb{P}_g\right), \tag{3}$$

where $p_d(\boldsymbol{x})$ and $p_g(\boldsymbol{x})$ are density functions of $\mathbb{P}_d$ and $\mathbb{P}_g$ with respect to $\mathbb{P}_m$.[1] The minimum of the JS divergence $D_{JS}\left(\mathbb{P}_d\|\mathbb{P}_g\right)$ equal to zero is attainable iff $\mathbb{P}_d = \mathbb{P}_g$, while zero mutual information indicates the independence between $\boldsymbol{x}$ and $s$, yielding $\mathbb{P}_{x,s}(\boldsymbol{x}, s) = \mathbb{P}_m(\boldsymbol{x})\mathbb{P}_s(s)$.

Exploiting the equality presented in (3), we proceed with proving Lemma 1, i.e., a series of JS divergence equalities.

## S2.2 Proof for Lemma 1

**Lemma 1.** *Let $s \in \{0, 1\}$ denote a random bit with uniform distribution $\mathbb{P}_s(s) = \frac{\delta[s] + \delta[s-1]}{2}$, where $\delta[s]$ is the Kronecker delta. Associating $s$ with $\boldsymbol{x}$, two joint distributions of $(\boldsymbol{x}, s)$ are constructed as*

$$\mathbb{P}_{\mathrm{x,s}}(\boldsymbol{x}, s) \triangleq \frac{\mathbb{P}_{\mathrm{d}}(\boldsymbol{x})\delta[s] + \mathbb{P}_{\mathrm{g}}(\boldsymbol{x})\delta[s-1]}{2}, \quad \mathbb{Q}_{\mathrm{x,s}}(\boldsymbol{x}, s) \triangleq \frac{\mathbb{P}_{\mathrm{g}}(\boldsymbol{x})\delta[s] + \mathbb{P}_{\mathrm{d}}(\boldsymbol{x})\delta[s-1]}{2}. \quad (4)$$

*Their JS divergence is equal to*

$$D_{\mathrm{JS}}\left(\mathbb{P}_{\mathrm{x,s}}\|\mathbb{Q}_{\mathrm{x,s}}\right) = D_{\mathrm{JS}}\left(\mathbb{P}_{\mathrm{d}}\|\mathbb{P}_{\mathrm{g}}\right). \quad (5)$$

*Taking (4) as the starting point and with $\boldsymbol{s}_l$ being a sequence of i.i.d. random bits of length $l$, the recursion of constructing the paired joint distributions of $(\boldsymbol{x}, \boldsymbol{s}_l)$*

$$\mathbb{P}_{\mathrm{x},\boldsymbol{s}_l}(\boldsymbol{x}, \boldsymbol{s}_l) \triangleq \mathbb{P}_{\mathrm{x},\boldsymbol{s}_{l-1}}(\boldsymbol{x}, \boldsymbol{s}_{l-1})\delta[s_l]/2 + \mathbb{Q}_{\mathrm{x},\boldsymbol{s}_{l-1}}(\boldsymbol{x}, \boldsymbol{s}_{l-1})\delta[s_l - 1]/2$$
$$\mathbb{Q}_{\mathrm{x},\boldsymbol{s}_l}(\boldsymbol{x}, \boldsymbol{s}_l) \triangleq \mathbb{Q}_{\mathrm{x},\boldsymbol{s}_{l-1}}(\boldsymbol{x}, \boldsymbol{s}_{l-1})\delta[s_l]/2 + \mathbb{P}_{\mathrm{x},\boldsymbol{s}_{l-1}}(\boldsymbol{x}, \boldsymbol{s}_{l-1})\delta[s_l - 1]/2 \quad (6)$$

*results into a series of JS divergence equalities for $l = 1, 2, \ldots, L$, i.e.,*

$$D_{\mathrm{JS}}\left(\mathbb{P}_{\mathrm{d}}\|\mathbb{P}_{\mathrm{g}}\right) = D_{\mathrm{JS}}\left(\mathbb{P}_{\mathrm{x},\boldsymbol{s}_1}\|\mathbb{Q}_{\mathrm{x},\boldsymbol{s}_1}\right) = \cdots = D_{\mathrm{JS}}\left(\mathbb{P}_{\mathrm{x},\boldsymbol{s}_L}\|\mathbb{Q}_{\mathrm{x},\boldsymbol{s}_L}\right). \quad (7)$$

*Proof.* Starting from the single bit $s$, the two joint distributions $\mathbb{P}_{\mathrm{x,s}}$ and $\mathbb{Q}_{\mathrm{x,s}}$ differ from each other by their opposite way of associating the bit $s \in \{0, 1\}$ with the data and synthetic samples. Their marginals with respect to $\boldsymbol{x}$ are identical and equal the mixture distribution $\mathbb{P}_{\mathrm{m}}$, being neither the data nor the model distribution, in contrast to the framework of [6].

The joint distribution $\mathbb{P}_{\mathrm{x,s}}$ has yielded the mutual information $I(\boldsymbol{x}; s)$ with the equality in (3). By analogy, we compute the mutual information $\tilde{I}(\boldsymbol{x}; s)$ between $\boldsymbol{x}$ and $s$ which follow $\mathbb{Q}_{\mathrm{x,s}}$ with the equality:

$$\tilde{I}(\boldsymbol{x}; s) = D_{\mathrm{JS}}\left(\mathbb{P}_{\mathrm{d}}\|\mathbb{P}_{\mathrm{g}}\right). \quad (8)$$

The combination of (3) and (8) leads to

$$D_{\mathrm{JS}}\left(\mathbb{P}_{\mathrm{d}}\|\mathbb{P}_{\mathrm{g}}\right) = \frac{I(\boldsymbol{x}; s) + \tilde{I}(\boldsymbol{x}; s)}{2}. \quad (9)$$

Rewriting mutual information as KL divergence yields:

$$D_{\mathrm{JS}}\left(\mathbb{P}_{\mathrm{d}}\|\mathbb{P}_{\mathrm{g}}\right) = \frac{D_{\mathrm{KL}}\left(\mathbb{P}_{\mathrm{x,s}}\|\mathbb{P}_{\mathrm{m}}\mathbb{P}_{\mathrm{s}}\right) + D_{\mathrm{KL}}\left(\mathbb{Q}_{\mathrm{x,s}}\|\mathbb{P}_{\mathrm{m}}\mathbb{P}_{\mathrm{s}}\right)}{2}, \quad (10)$$

where $\mathbb{P}_{\mathrm{m}}$ and $\mathbb{P}_{\mathrm{s}}$ are the common marginals of $\mathbb{P}_{\mathrm{x,s}}$ and $\mathbb{Q}_{\mathrm{x,s}}$ with respect to $\boldsymbol{x}$ and $s$. By further identifying

$$\mathbb{P}_{\mathrm{m}}(\boldsymbol{x})\mathbb{P}_{\mathrm{s}}(s) = \frac{\mathbb{P}_{\mathrm{x,s}}(\boldsymbol{x}, s) + \mathbb{Q}_{\mathrm{x,s}}(\boldsymbol{x}, s)}{2} \quad (11)$$

and plugging it into (10), we finally reach to

$$D_{\mathrm{JS}}\left(\mathbb{P}_{\mathrm{d}}\|\mathbb{P}_{\mathrm{g}}\right) = D_{\mathrm{JS}}\left(\mathbb{P}_{\mathrm{x,s}}\|\mathbb{Q}_{\mathrm{x,s}}\right) \quad (12)$$

by the definition of JS divergence.

It is worth noting that the equivalence holds even if the feasible solution set of $\mathbb{P}_{\mathrm{g}}$ determined by $G$ does not include the data distribution $\mathbb{P}_{\mathrm{d}}$. This is of practical interest as it is often difficult to guarantee the fulfillment of such premise when modeling $G$ by means of neural networks.

Replacing the data and model distributions $\mathbb{P}_{\mathrm{d}}$ and $\mathbb{P}_{\mathrm{g}}$ respectively with $\mathbb{P}_{\mathrm{x,s}}$ and $\mathbb{Q}_{\mathrm{x,s}}$, we can systematically add a new bit with the same derivation as above. Repeating this procedure $L$ times eventually yields the recursively constructed $\{\mathbb{P}_{\mathrm{x},\boldsymbol{s}_l}, \mathbb{Q}_{\mathrm{x},\boldsymbol{s}_l}\}_{l=1,\ldots,L}$ followed by a sequence of JS divergence equalities

$$D_{\mathrm{JS}}\left(\mathbb{P}_{\mathrm{d}}\|\mathbb{P}_{\mathrm{g}}\right) = \cdots = D_{\mathrm{JS}}\left(\mathbb{P}_{\mathrm{x},\boldsymbol{s}_{l-1}}\|\mathbb{Q}_{\mathrm{x},\boldsymbol{s}_{l-1}}\right) = \cdots = D_{\mathrm{JS}}\left(\mathbb{P}_{\mathrm{x},\boldsymbol{s}_L}\|\mathbb{Q}_{\mathrm{x},\boldsymbol{s}_L}\right). \quad (13)$$

$\square$

## S2.3 Proof for Theorem 1

**Theorem 1.** *The min-max optimization problem of GANs [8] is equivalent to*

$$\min_G \max_D \mathbb{E}_{\mathbb{P}_{x,s_l}} \left\{ \log \left[ D(\boldsymbol{x}, \boldsymbol{s}_l) \right] \right\} + \mathbb{E}_{\mathbb{Q}_{x,s_l}} \left\{ \log \left[ 1 - D(\boldsymbol{x}, \boldsymbol{s}_l) \right] \right\} \quad \forall l \in \{1, 2, \ldots, L\}, \qquad (14)$$

*where the two joint distributions, i.e., $\mathbb{P}_{x,s_l}$ and $\mathbb{Q}_{x,s_l}$, are defined in (6) and the function $D$ maps $(\boldsymbol{x}, \boldsymbol{s}_l) \in \mathcal{X} \times \{0,1\}^l$ onto $[0,1]$. For a fixed $G$, the optimal $D$ is*

$$D^*(\boldsymbol{x}, \boldsymbol{s}_l) = \frac{\mathbb{P}_{x,s_l}(\boldsymbol{x}, \boldsymbol{s}_l)}{\mathbb{P}_{x,s_l}(\boldsymbol{x}, \boldsymbol{s}_l) + \mathbb{Q}_{x,s_l}(\boldsymbol{x}, \boldsymbol{s}_l)} = \frac{\mathbb{P}_d(\boldsymbol{x})}{\mathbb{P}_d(\boldsymbol{x}) + \mathbb{Q}_d(\boldsymbol{x})}, \qquad (15)$$

*whereas the attained inner maximum equals $D_{\mathrm{JS}}\left(\mathbb{P}_{x,s_l} \| \mathbb{Q}_{x,s_l}\right) = D_{\mathrm{JS}}\left(\mathbb{P}_d \| \mathbb{P}_g\right)$ for $l = 1, 2, \ldots, L$.*

*Proof.* Analogous to the proofs for GANs [8, Sec.4], we can construct a binary classification task for computing JS divergences, i.e.,

$$D_{\mathrm{JS}}\left(\mathbb{P}_{x,s_l} \| \mathbb{Q}_{x,s_l}\right) = \max_D \mathbb{E}_{\mathbb{P}_{x,s_l}} \left\{ \log \left[ D(\boldsymbol{x}, \boldsymbol{s}_l) \right] \right\} + \mathbb{E}_{\mathbb{Q}_{x,s_l}} \left\{ \log \left[ 1 - D(\boldsymbol{x}, \boldsymbol{s}_l) \right] \right\} \quad \forall l, \qquad (16)$$

where the optimal $D^*$ equals

$$D^*(\boldsymbol{x}, \boldsymbol{s}_l) = \frac{\mathbb{P}_{x,s_l}(\boldsymbol{x}, \boldsymbol{s}_l)}{\mathbb{P}_{x,s_l}(\boldsymbol{x}, \boldsymbol{s}_l) + \mathbb{Q}_{x,s_l}(\boldsymbol{x}, \boldsymbol{s}_l)} \overset{(a)}{=} \frac{\mathbb{P}_d(\boldsymbol{x})}{\mathbb{P}_d(\boldsymbol{x}) + \mathbb{Q}_d(\boldsymbol{x})}. \qquad (17)$$

The equality $(a)$ in above is based on the recursive construction of $\mathbb{P}_{x,s_l}$ and $\mathbb{Q}_{x,s_l}$ from $\mathbb{P}_d$ and $\mathbb{P}_g$.

The equalities in (5) imply that for any given pair $(\mathbb{P}_d, \mathbb{P}_g)$ the correspondingly constructed joint distribution pair $(\mathbb{P}_{x,s_l}, \mathbb{Q}_{x,s_l})$ yields the same JS divergence. For this reason, we can use the two JS divergences interchangeably as the objective function while optimizing $\mathbb{P}_g$, yielding

$$\min_G \max_D \mathbb{E}_{\mathbb{P}_d} \left\{ \log \left[ D(\boldsymbol{x}) \right] \right\} + \mathbb{E}_{\mathbb{P}_g} \left\{ \log \left[ 1 - D(\boldsymbol{x}) \right] \right\}$$

$$\equiv \min_G \max_D \mathbb{E}_{\mathbb{P}_{x,s_l}} \left\{ \log \left[ D(\boldsymbol{x}, \boldsymbol{s}_l) \right] \right\} + \mathbb{E}_{\mathbb{Q}_{x,s_l}} \left\{ \log \left[ 1 - D(\boldsymbol{x}, \boldsymbol{s}_l) \right] \right\} \quad \forall l. \qquad (18)$$

$\square$

## S2.4 Generalization of Lemma 1

In this work, we base the development of PA on Lemma 1 and Theorem 1. From a broader perspective, the random bits $\boldsymbol{s}$ can be any generic random variables applicable for generative modelling.

**Proposition 1.** *Let $\boldsymbol{s}$ denote a random variable with two unequal distributions $\mathbb{P}_{s,a}$ and $\mathbb{P}_{s,b}$. Together with the two distributions $\mathbb{P}_d$ and $\mathbb{P}_g$ of $\boldsymbol{x}$, two joint distributions are constructed as follows:*

$$\begin{aligned} \mathbb{P}_{x,s}(\boldsymbol{x}, \boldsymbol{s}) &= \frac{\mathbb{P}_d(\boldsymbol{x})\mathbb{P}_{s,a}(\boldsymbol{s}) + \mathbb{P}_g(\boldsymbol{x})\mathbb{P}_{s,b}(\boldsymbol{s})}{2} \\ \mathbb{Q}_{x,s}(\boldsymbol{x}, \boldsymbol{s}) &= \frac{\mathbb{P}_d(\boldsymbol{x})\mathbb{P}_{s,b}(\boldsymbol{s}) + \mathbb{P}_g(\boldsymbol{x})\mathbb{P}_{s,a}(\boldsymbol{s})}{2} \end{aligned} . \qquad (19)$$

*The mutual information $I(\boldsymbol{x}; \boldsymbol{s})$ and $\tilde{I}(\boldsymbol{x}; \boldsymbol{s})$, with respect to $\mathbb{P}_{x,s}$ and $\mathbb{Q}_{x,s}$, are minimized to zero if $\mathbb{P}_d = \mathbb{P}_g$. When $\mathbb{P}_{s,a}$ and $\mathbb{P}_{s,b}$ have non-overlapped supports, the JS divergence between $\mathbb{P}_{x,s}$ and $\mathbb{Q}_{x,s}$ equals the JS divergence between $\mathbb{P}_d$ and $\mathbb{P}_g$, i.e., $D_{\mathrm{JS}}\left(\mathbb{P}_d \| \mathbb{P}_g\right) = D_{\mathrm{JS}}\left(\mathbb{P}_{x,s} \| \mathbb{Q}_{x,s}\right)$.*

*Proof.* The mutual information between $\boldsymbol{x}$ and $\boldsymbol{s}$ is minimized and equal zero if they are independent. Under the condition $\mathbb{P}_{s,a} \neq \mathbb{P}_{s,b}$, the two joint distributions $\mathbb{P}_{x,s}$ and $\mathbb{Q}_{x,s}$ become factorizable if $\mathbb{P}_d = \mathbb{P}_g$. Analogous to the proof of Lemma 1, the JS divergence between $\mathbb{P}_{x,s}$ and $\mathbb{Q}_{x,s}$ equals the mean of $I(\boldsymbol{x}; \boldsymbol{s})$ and $\tilde{I}(\boldsymbol{x}; \boldsymbol{s})$, i.e.,

$$D_{\mathrm{JS}}\left(\mathbb{P}_{x,s} \| \mathbb{Q}_{x,s}\right) = \frac{I(\boldsymbol{x}; \boldsymbol{s}) + \tilde{I}(\boldsymbol{x}; \boldsymbol{s})}{2}. \qquad (20)$$

Expressing mutual information as KL divergence plus the condition that $\mathbb{P}_{s,a}$ and $\mathbb{P}_{s,b}$ have non-overlapped supports, we reach to (3) for both $I(\boldsymbol{x}; \boldsymbol{s})$ and $\tilde{I}(\boldsymbol{x}; \boldsymbol{s})$ and thereby conclude the proof. $\square$

Lemma 1 is a special case of Proposition 1, namely, $\mathbb{P}_{s,a}(s) = \delta[s]$ and $\mathbb{P}_{s,b}(s) = \delta[s-1]$.

# S3   Implementation Details of PA-GAN

## S3.1   Input and Feature Space Augmentation

As being presented in Sec. 3.2, we spatially replicate each augmentation bit and perform depth concatenation with the input $\boldsymbol{x}$ or its learned feature maps at the intermediate hidden layers. After concatenation along the channel axis, the input layer or the hidden layer then process such augmented input. For instance, in the case of a convolutional layer, it processes the augmented input as

$$\mathrm{conv}(\phi(\boldsymbol{x}), s_1, \ldots, s_l) = \mathrm{conv}(\phi(\boldsymbol{x})) + \sum_l \mathrm{conv}(s_l) \tag{21}$$

where the kernel width/height, stride and padding type used for filtering the augmentation bits are the same as that of $\phi(\boldsymbol{x})$.[2] Depending on the augmentation space, here $\phi(\boldsymbol{x})$ collectively denotes either the input $\boldsymbol{x}$ or its feature maps. When spectral normalization is in use, the power method is applied to estimate the largest singular value of the filter matrix that processes the augmented input. In case of augmenting the input to a residual block, the augmentation bits are passed along with $\boldsymbol{x}$ or its feature maps into the first convolutional layer in the main branch as well as into the shortcut connection. We bypass the shortcut connection if it is an identity mapping.

When progression scheduling increases the augmentation level, a new set of filter coefficients are instantiated to process the new augmentation bit according to (21). They are initialized by random Gaussian variables with the mean and variance computed from the existing filter coefficients for $\phi(\boldsymbol{x})$. Before filtering, each augmentation bit can be additionally modulated by two trainable parameters $\{\lambda_l, \beta_l\}$. The scaling parameter $\lambda_l$ is initialized with the mean value of the previous ones, where the first one, i.e., $\lambda_1$, is initialized as one. The offset parameters $\{\beta_l\}$ are always initialized as zeros.

## S3.2   Mini-batch Discrimination

Each mini-batch is constructed with the same number of real data samples, synthetic samples and bit sequences. Each bit sequence is randomly sampled and associated with one real and one synthetic sample. Based on the checksums of the formed pairs, we can decide their correct class and feed it into the discriminator to compute the cross-entropy loss. This way of generating $(\boldsymbol{x}, \boldsymbol{s})$ guarantees a balanced number of TRUE/FAKE samples, forming the two mini-batches $\mathcal{B}_{\mathrm{tr}}$ and $\mathcal{B}_{\mathrm{fk}}$.

## S3.3   Warm-up Phase of Progression

At the beginning of the new augmentation level the discriminator is ignorant about this disruptive change and as the bit $s = 1$ flips the reference label it will lead to about $50\%$ discriminator errors in one mini-batch. Aiming at a smooth transition from the current augmentation level to the new one, here we introduce two warm-up mechanisms that are usable when the discriminator exhibits deficiency in timely coping with the new augmentation level.

The first mechanism instantiates an Adam optimizer, independent of the ones for $D$ and $G$, to solely train the newly introduced weights right after progressing to the new level. It takes the $D$ loss and can use the same learning hyperparameters as those of the $D$ optimizer. After multiple iterations (e.g., 1k), we continue with the original alternation between the $D$ and $G$ optimizer, where the new weights together with the existing ones of the discriminator network are handled by the $D$ optimizer.

According to Lemma 1, the augmentation bits shall follow a uniform distribution, i.e., $\mathbb{P}(s = 1) = p$ and $\mathbb{P}(s = 0) = 1 - p$ with $p = 0.5$. As the new augmentation bit taking on the value one causes discriminator errors, the second mechanism temporally adopts a non-uniform distribution when kicking off a new augmentation level. Namely, we can on purpose create more 0s than 1s by linearly increasing $p$ from 0 and 0.5 within a given number of iterations, e.g., 5k.

## S3.4   Loss Functions

In this work, we experimented of using PA with the following loss functions of GANs.

**Non-saturating (NS) loss.** The cross-entropy loss for $D$ is given as

$$\min_D -\mathbb{E}_{\mathbb{P}_{x,s_l}} \left\{ \log D(\boldsymbol{x}, \boldsymbol{s}_l) \right\} - \mathbb{E}_{\mathbb{Q}_{x,s_l}} \left\{ \log \left[ 1 - D(\boldsymbol{x}, \boldsymbol{s}_l) \right] \right\}. \tag{22}$$

Since both distribution $\mathbb{P}_{x,s_l}$ and $\mathbb{Q}_{x,s_l}$ involve synthetic samples, the non-saturating (NS) loss for $G$ [8] is reformulated as

$$\min_G -\mathbb{E}_{\mathbb{Q}_{x,s_l}} \left\{ \log D(\boldsymbol{x}, \boldsymbol{s}_l) \right\} - \mathbb{E}_{\mathbb{P}_{x,s_l}} \left\{ \log \left[ 1 - D(\boldsymbol{x}, \boldsymbol{s}_l) \right] \right\}. \tag{23}$$

During training, the two expectations are approximated by averaging over the samples in the TRUE/FAKE mini-batches $\mathcal{B}_{\text{tr}}$ and $\mathcal{B}_{\text{fk}}$, which construction is discussed in Sec. S3.2.

**Hinge loss.** Instead of cross-entropy loss, $D$ can also be trained using the hinge loss

$$\min_D \mathbb{E}_{\mathbb{P}_{x,s_l}} \left\{ \max \left[ 0, 1 - D(\boldsymbol{x}, \boldsymbol{s}_l) \right] \right\} + \mathbb{E}_{\mathbb{Q}_{x,s_l}} \left\{ \max \left[ 0, 1 + D(\boldsymbol{x}, \boldsymbol{s}_l) \right] \right\}. \tag{24}$$

Accordingly, the $G$ loss is adapted to

$$\min_G \mathbb{E}_{\mathbb{P}_{x,s_l}} \left\{ D(\boldsymbol{x}, \boldsymbol{s}_l) \right\} - \mathbb{E}_{\mathbb{Q}_{x,s_l}} \left\{ D(\boldsymbol{x}, \boldsymbol{s}_l) \right\}. \tag{25}$$

**WGAN-GP.** In the main paper, we have focused on generative modeling with JS divergence. It is also possible to interchange the JS divergence with the Wasserstein distance and then cast GAN training into WGAN-GP training [1]. Wasserstein distance is weaker than JS divergence and $D$ termed critic in WGAN no longer solves the classification task. So, we alternatively exploit the stochastic model averaging role of the augmentation bits rather than their regularization role.

Briefly, with the Kantorovich-Rubinstein duality, minimizing the Wasserstein distance between $\mathbb{P}_d$ and $\mathbb{P}_g$ is transformed into the following two-player game

$$\min_G \max_D \mathbb{E}_{\mathbb{P}_{x,s_l}} \left\{ D(\boldsymbol{x}, \boldsymbol{s}_l) \right\} - \mathbb{E}_{\mathbb{Q}_{x,s_l}} \left\{ D(\boldsymbol{x}, \boldsymbol{s}_l) \right\}. \tag{26}$$

Ideally, $D$ in the context of WGAN should be 1-Lipschitz continuous. As a pragmatic relaxation on this constraint, a gradient penalty (GP) [9] is commonly added to the objective function when optimizing $D$.

Within the same mini-batch of $\boldsymbol{x} \sim \mathbb{P}_d$ and $\boldsymbol{x} \sim \mathbb{P}_g$, we draw $M$ mini-batches $\boldsymbol{s} \sim \mathbb{P}_s$ of the same size. Combining each of them with the data and synthetic samples, we create $M$ mini-batches for approximating the expectations in the objective function

$$\mathbb{E}_{\mathbb{P}_{x,s_l}} \left\{ D(\boldsymbol{x}, \boldsymbol{s}_l) \right\} - \mathbb{E}_{\mathbb{Q}_{x,s_l}} \left\{ D(\boldsymbol{x}, \boldsymbol{s}_l) \right\}$$

$$\approx L_m \triangleq \frac{1}{|\mathcal{B}_{\text{tr},m}|} \sum_{(\boldsymbol{x}, \boldsymbol{s}_l) \in \mathcal{B}_{\text{tr},m}} D(\boldsymbol{x}, \boldsymbol{s}_l) - \frac{1}{|\mathcal{B}_{\text{fk},m}|} \sum_{(\boldsymbol{x}, \boldsymbol{s}_l) \in \mathcal{B}_{\text{fk},m}} D(\boldsymbol{x}, \boldsymbol{s}_l), \quad m = 1, \ldots, M. \tag{27}$$

The critic $D$ of WGAN-GP is trained to maximize the averaged loss $L_m$ across the $M$ mini-batches, making use of stochastic model averaging. The generator $G$ is then trained to minimize the maximum of $\{L_m\}$, $m = 1, \ldots, M$, i.e. picking the best performing case of the critic, as a good quality of the critic $D$ is important to the optimization process of $G$ in the context of WGAN. With single bit augmentation of `PA (feat)` and two draws per minibatch, we can improve WGAN-GP of `SN DCGAN` on CIFAR10 from 25.0 to 23.9 FID. Here, we boost the diversity of the two draws by choosing them with opposite checksums.

## S4   Additional Ablation Studies

In this section, we provide additional ablation studies of PA. Complementary to Table 1 in Sec. 4.1., an ablation study on the choice of augmentation space is conducted in Sec. S4.1, evaluating PA across input, low- and high-level feature space augmentation. One important finding in Sec. 4.2. of the main paper is that dropout and PA are complementary and mutually beneficial. In Sec. S4.2, we report our detailed investigation on the dropout regularization followed by evaluation of its combination with PA across the datasets and architectures. The two time-scale update rule (TTUR) [10], updating the discriminator and generator with different learning rates, is notoriously helpful to stabilize GAN training. In Sec. S4.3, we examine the performance of PA under different TTURs and then compare it with the adaptive learning rate.

**Table S1:** Median FIDs of input and feature space augmentation across five random runs. We experiment with augmenting input and features at different intermediate layers, e.g. $\texttt{feat}_{N/4}$ denotes layer with the spatial dimension $N/4$, where $N$ is the input image dimension.

| Method | Dataset | PA | | | | |
|---|---|---|---|---|---|---|
| | | ✗ | input (N) | $\texttt{feat}_{N/2}$ | $\texttt{feat}_{N/4}$ | $\texttt{feat}_{N/8}$ |
| SN DCGAN - NS Loss | CIFAR10 | 26.0 | **22.2** | 22.8 | 22.7 | 22.6 |
| | CELEBA-HQ | 24.3 | 20.8 | 19.6 | **18.8** | **18.8** |
| SA GAN (sBN) - Hinge Loss | CIFAR10 | 18.8 | **16.1** | 16.3 | 16.3 | - |
| | CELEBA-HQ | 17.8 | **15.4** | **15.4** | 16.4 | 15.8 |

**Table S2:** Median FIDs (across five random runs) of $\texttt{Dropout}$ and $\texttt{SpatialDropout}$ applied on the input layer or intermediate layers with different keep rates on CIFAR10 using SN DCGAN.

| Keep rate | Dropout | | | | SpatialDropout | | | |
|---|---|---|---|---|---|---|---|---|
| | input (N) | $\texttt{feat}_{N/2}$ | $\texttt{feat}_{N/4}$ | $\texttt{feat}_{N/8}$ | input (N) | $\texttt{feat}_{N/2}$ | $\texttt{feat}_{N/4}$ | $\texttt{feat}_{N/8}$ |
| 1.0 | 26.0 | | | | | | | |
| 0.95 | 25.5 | 25.6 | 24.1 | 25.3 | 26.0 | 25.3 | 24.9 | 26.0 |
| 0.9 | 26.4 | 25.1 | 23.4 | 24.6 | 26.2 | 25.3 | 24.0 | 25.8 |
| 0.7 | 28.0 | 25.6 | **22.1** | 24.4 | 27.6 | 26.1 | 23.4 | 25.3 |
| 0.5 | 27.1 | 25.9 | 23.1 | 24.0 | 29.7 | 26.9 | 24.1 | 25.4 |
| 0.3 | 27.7 | 25.6 | 22.4 | 24.6 | 31.3 | 28.8 | 24.6 | 25.8 |
| 0.1 | 32.3 | 28.6 | 24.3 | 23.9 | 45.7 | 37.7 | 28.8 | 25.8 |

## S4.1 Ablation Study on Augmentation Space

In the main paper, in Table 1 of Sec. 4.1. we reported the FID scores achieved by PA, by augmenting either the input - PA ($\texttt{input}$), or its features with spatial dimension $N/8$ - PA ($\texttt{feat}_{N/8}$), where $N$ is the input image dimension (see Sec. S8 for the detailed configuration). Here, we further perform the ablation study on the choice of the augmentation space across two datasets (CIFAR10 and CELEBA-HQ) and two architectures (SN DCGAN and SA GAN). From Table S1, we observe the stable performance improvement across all configurations, inline with Table 1 of the main paper. The performance difference across different feature space augmentations is generally small (less than one FID point).

## S4.2 Ablation Study on Dropout and its Combination with PA

In Sec. 4.2. of the main paper, we have shown the effectiveness of using dropout, particularly, in combination with the proposed PA. In this part we report further ablations for both techniques.

We start from applying dropout at the input layer and different intermediate layers. Note that, in contrast to dropout, we apply PA directly on the input and not on the input layer. In addition, we experiment with different keep rates of the dropout, i.e. $\{0.1, 0.3, 0, 5, 0.7, 0.9, 0.95\}$. Table S2 reports the FID scores achieved with different dropout configurations. In contrast to PA (see Table S1 or Table 1 in the main paper), the performance of dropout is very dependent on the applied layer and the selected keep rate. The feature space with the spatial dimension $N/4$ together with the keep rate 0.7 is the best performing setting on CIFAR10 with SN DCGAN.

We further note that the binary dropout mask is independently drawn for each entry of the input or intermediate layer outputs (each convolution feature map activation is "dropped-out" independently). In addition, we also experiment with the *spatial* dropout ($\texttt{SpatialDropout}$) [21], which randomly drops the entire feature maps instead of individual elements. The results in Tables S2 show that the entry-wise dropout outperforms the spatial dropout in the context of GAN training, i.e., FID 22.1 vs. 23.4. Therefore we only consider the entry-wise dropout for comparison with PA in the main paper.

In Table 3 of the main paper, we have successfully combined dropout at its best setting with PA on CIFAR10 with SN DCGAN and SA GAN. Table S3 and S4 additionally report the FID improvements where dropout is applied at different intermediate layers and keep rates. In all configurations, PA

**Table S3:** Median FIDs (across five random runs) of PA together with dropout applied on different intermediate layers with the keep rate $0.7$ and on CIFAR10 and CELEBA-HQ.

| Method | Dataset | PA | GAN | -Dropout [19] $\text{feat}_{N/8}$ | $\text{feat}_{N/4}$ | $\text{feat}_{N/2}$ | $\overline{\Delta\text{PA}}$ |
|---|---|---|---|---|---|---|---|
| SN DCGAN - NS Loss | CIFAR10 | ✗ | 26.0 | 24.4 | 22.1 | 25.6 | 2.0 |
| | | $\text{feat}_{N/8}$ | 22.6 | 21.3 | **20.6** | 22.5 | 1.1 |
| SA GAN (sBN) - Hinge Loss | | ✗ | 18.8 | — | 16.2 | 17.1 | 2.2 |
| | | $\text{feat}_{N/4}$ | 16.3 | — | **15.6** | 15.7 | 0.7 |
| SN DCGAN - NS Loss | CELEBA-HQ | ✗ | 24.3 | — | 24.0 | — | 0.3 |
| | | $\text{feat}_{N/8}$ | 18.8 | — | **18.1** | — | 0.7 |
| | | $\overline{\Delta\text{PA}}$ | 3.8 | 3.1 | 2.7 | 2.3 | |

**Table S4:** Median FIDs (across five random runs) of PA together with dropout applied on different intermediate layers and keep rates on CIFAR10 with SN DCGAN.

| | Dropout $\text{PA}(\text{feat}_{N/8})$ | $\text{input}(\text{N})$ ✗ | ✓ | $\text{feat}_{N/2}$ ✗ | ✓ | $\text{feat}_{N/4}$ ✗ | ✓ | $\text{feat}_{N/8}$ ✗ | ✓ |
|---|---|---|---|---|---|---|---|---|---|
| | 0.9 | 26.4 | 22.6 | 25.1 | 21.9 | 23.4 | 21.2 | 24.6 | 21.6 |
| Keep Rate | 0.7 | 28.0 | 22.9 | 25.6 | 21.3 | 22.1 | **20.6** | 24.4 | 22.5 |
| | 0.5 | 27.1 | 23.1 | 25.9 | 22.3 | 23.1 | 21.2 | 24.0 | 22.1 |
| $\Delta\text{PA}$ | | 4.5 | | 3.7 | | 1.9 | | 2.3 | |

**Table S5:** Median FIDs (across five random runs) of different learning rates (TTURs) on CIFAR10 with SN DCGAN. Italic and bold denotes the best FIDs w/o and with PA respectively, underline denotes the default learning rate setting of SN DCGAN.

| $\text{lr}_g$ \\ $\text{lr}_d$ | PA ($\text{feat}_{N/8}$) | $10^{-4}$ | $2 \times 10^{-4}$ | $4 \times 10^{-4}$ | $10^{-3}$ | $\overline{\Delta\text{PA}}$ |
|---|---|---|---|---|---|---|
| $10^{-4}$ | ✗ | 27.0 | 25.8 | *25.3* | 27.0 | 3.5 |
| | ✓ | 23.3 | **22.2** | 22.6 | 22.9 | |
| $2 \times 10^{-4}$ | ✗ | 26.7 | 26.0 | 26.2 | 27.2 | 3.1 |
| | ✓ | 24.8 | 22.6 | 22.3 | 24.0 | |
| $4 \times 10^{-4}$ | ✗ | 28.7 | 26.1 | 26.3 | 28.2 | 3.6 |
| | ✓ | 24.7 | 23.3 | 22.9 | 24.2 | |
| $10^{-3}$ | ✗ | 28.5 | 27.0 | 26.4 | 27.4 | 2.9 |
| | ✓ | 25.7 | 23.6 | 23.4 | 25.0 | |

provides complementary gains. Note that, for CELEBA-HQ Dropout alone in Table S3 only has a marginal performance improvement over the baseline, whereas its combination with PA leads to larger performance boost. Overall, Table S3, S4 plus Table 3 in the main paper confirms the effectiveness of exploiting both techniques. Adding PA is beneficial independent of the dropout settings (keep rate and applied layer), it helps to reduce the FID sensitivity to the dropout hyperparameter choice.

### S4.3 Ablation Study on Learning Rates

Table S5 compares the performance achieved by using different learning rate configurations. The improvement achieved by PA is consistent across different settings ($\sim 3$ FID points), showing its robustness to different update rules. Compared to the best performing TTUR, PA reduces the FID faster over iterations (see Figure S1) without requiring extra hyperparameter search for the best update rule.

Table S5 has also shown a stable FID performance of SN DCGAN with the generator learning rate $\text{lr}_g = 2 \times 10^{-4}$ and the discriminator learning rate $\text{lr}_d \in \{10^{-4}, 2 \times 10^{-4}, 4 \times 10^{-4}\}$. With this identification, we fix $\text{lr}_g = 2 \times 10^{-4}$ and reuse the progression scheduling to adaptively reduce $\text{lr}_d$ from $4 \times 10^{-4}$ to $10^{-4}$ with the learning rate decay of $0.8$ (in our experiments the best performing learning rate decay among $\{0.99, 0.95, 0.9, 0.8, 0.7\}$). Figure S1 shows the effectiveness

**Figure S1:** FID learning curves (mean FIDs with one standard deviation across five random runs) of `PA`, `TTUR`, adaptive learning rate and `Dropout` on CIFAR10 with `SN DCGAN`.

of progression scheduling in assisting both the learning rate adaptation and progressive augmentation for an improved performance. `PA` outperforms learning rate adaptation as well as the tuned `TTUR` [10] , i.e. FID 22.6 vs. 24.0 vs. 25.3. Its combination with `Dropout` delivers the best performance in this experiment, i.e., 20.6.

## S5    Effectiveness of PA as a Regularizer

Here we exploit progressive augmentation on a toy classification task to empirically illustrate its regularization benefits discussed in Sec. 3 of the main paper. Specifically, we focus on binary classification task taking the alike Cat and Dog images from CIFAR10 [12], which represent the TRUE (real) and FAKE (synthetic) data samples, and train the discriminator network of SN DCGAN with the cross-entropy loss to tell them apart. Figure S2 depicts the discriminator loss ($D$ loss) behaviour over iterations on the training and test sets. It shows that the discriminator very quickly becomes over-confident on the training set and that overfitting takes place after 1k iterations.

In order to regularize the discriminator we exploit the proposed progressive augmentation (PA), augmenting either the input - `PA (input)`, or its features with spatial dimension $N/8$ - `PA (feat`$_{N/8}$`)`, where $N$ is the input image dimension. For a comparison purpose, we also experiment with the `Dropout` [19] regularization applied on `feat`$_{N/4}$ layer with the keep rate $0.7$ (the best performing rate in our experiments). Both techniques resort to random variables for regularization. The former randomly removes features, while the latter augments them with additional random bits and adjusts accordingly the class label. In contrast to `Dropout`, PA exhibits a long lasting regularization effect by means of progression. Each rise of $D$ loss coinciding with an iteration at which the augmentation level increases (every 2k iterations) and then gradually reduces after the discriminator timely adapts to the new bit. At the level one augmentation, both `PA (input)` and `PA (feat`$_{N/8}$`)` start from the similar overfitting stage. Combining the bit $s$ directly with high-level features eases checksum computation. As a result, the $D$ loss of `PA (feat`$_{N/8}$`)` reduces faster, but making its future task more difficult due to overfitting to the previous augmentation level. On the other hand, `PA (input)` let the bits pass through all layers, and thus its adaptation to augmentation progression improves over iterations. In the end, both `PA (input)` and `PA (feat`$_{N/8}$`)` lead to similar regularization effect. In addition, we compare PA with the `Reinit.` baseline, where every 2k iterations all weights are reinitialized with Xavier initialization [7]. Compared to PA, using `Reinit.` strategy leads to longer adaptation time (the $D$ loss decay is much slower), potentially providing non-informative signal to the generator and thus slowing down the training.

In Figure S3 we explore the stochastic nature of `Dropout` and PA. Each realization of the dropout mask or the augmentation bit sequence $s$ changes the loss function landscape, varying its gradient with respect to the synthetic sample (i.e. the Dog class in this case). With the same experimental setup, we now assess the correlation of the gradients based on the first four eigenvalues of their correlation matrix - $\lambda_i$, $i = 0, \ldots, 3$, i.e. computing the averaged square roots of their ratios $\bar{\gamma} \triangleq \frac{1}{3} \sum_{i=1}^{3} \sqrt{\lambda_0/\lambda_i}$. Figure S3 depicts the histograms of $\bar{\gamma}$ among $10^3$ instances. PA has more instances with smaller $\bar{\gamma}$ in comparison to `Dropout`, indicating a more diverse set of gradients, exploitable by the generator to approach the data distribution. In contrast to `Dropout`, in PA the

**Figure S2:** Behaviour of the discriminator loss ($D$ loss) with and w/o `PA` and in comparison to `Dropout`, using the $D$ architecture of SN DCGAN. See Sec. S5 for details.

**Figure S3:** Histograms of averaged square roots of eigenvalue ratios computed from gradient correlation matrices for `PA` and `Dropout`. Smaller correlation values indicate a more diverse set of gradients exploitable by the generator to approach the data distribution. See Sec. S5 for details.

**Table S6:** KID/IS improvements with `PA` across different datasets and network architectures, in accordance with Table 1 in the main paper.

| | | KID | | | | | IS | | |
|---|---|---|---|---|---|---|---|---|---|
| Method | PA | F-MNIST | CIFAR10 | CELEBA-HQ | T-ImageNet | $\overline{\Delta\text{PA}}$ | CIFAR10 | T-ImageNet | $\overline{\Delta\text{PA}}$ |
| SN DCGAN | ✗ | 0.004 | 0.016 | 0.011 | - | | 7.6 | - | |
| NS Loss | input | **0.002** | **0.013** | 0.007 | - | 0.003 | **7.8** | - | 0.2 |
| [16] | feat | **0.002** | **0.013** | **0.005** | - | | **7.8** | - | |
| SA GAN (sBN) | ✗ | - | 0.011 | 0.006 | 0.035 | | 8.4 | 8.8 | |
| Hinge Loss | input | - | **0.008** | **0.004** | **0.033** | 0.002 | **8.7** | 9.1 | 0.3 |
| [22] | feat | - | 0.009 | **0.004** | **0.033** | | 8.6 | **9.2** | |

augmentation random bits determine the target class in binary classification and the discriminator is trained to comprehend $s$ together with $x$, leading to the richer loss function landscape. Between input and feature space augmentation, the former yields more diverse gradients than the latter as $s$ is passed through all layers.

## S6 Exemplar Synthetic Samples

Figure S4 shows a set of synthetic samples that are outcomes of GAN training with and without `PA`. `PA` not only improves sample quality and variation, but also sensibly navigates the image manifold through latent space interpolation.

## S7 Evaluation with Other Performance Measures

In addition to FID, here we measure the quality of synthetic samples by means of kernel inception distance (KID) [2] and inception score (IS) [20], see Tables S6 and S7 which correspond to Tables 1 and 3 in the main paper. The evaluation framework setup is the same as that with FID and follows [15, 13]. For Fashion-MNIST and CELEBA-HQ, IS computed from the pre-trained Inception network is not meaningful and thus omitted. Overall, the obtained results show consistent observations with those that are made in Sec. 4 of the main paper based on the FID measure.

(a) SN DCGAN

(b) SN DCGAN with PA

(c) SA GAN (sBN)

(d) SA GAN (sBN) with PA

**Figure S4:** Synthetic samples from training SN GAN on Fashion-MNIST ($28 \times 28$) and SA GAN (sBN) on CELEBA-HQ ($128 \times 128$) with and without using PA. In all cases, i.e., (a), (b), (c) and (d), the eight images per row are generated through polar-interpolation between two randomly sampled $z_1$ and $z_2$.

**Table S7:** KIDs/ISs of PA, different regularization techniques and their combinations on CIFAR10, in according with Table 3 in the main paper.

KID

| Method | PA | GAN | -Label smooth. [18] | -GP [9] | -GP$_{\text{zero-cent}}$ [17] | -Dropout [19] | -SS [5] | $\overline{\Delta\text{PA}}$ |
|---|---|---|---|---|---|---|---|---|
| SN DCGAN | ✗ | 0.016 | 0.016 | 0.018 | 0.017 | 0.013 | — | 0.003 |
| NS Loss | feat | 0.013 | 0.014 | 0.014 | 0.014 | **0.012** | — | |
| SA GAN (sBN) | ✗ | 0.011 | — | 0.010 | 0.010 | 0.008 | 0.008 | 0.001 |
| Hinge Loss | feat | 0.009 | — | 0.008 | 0.008 | 0.008 | **0.007** | |
| | $\overline{\Delta\text{PA}}$ | 0.003 | 0.002 | 0.003 | 0.003 | 0.001 | 0.001 | |

IS

| Method | PA | GAN | -Label smooth. [18] | -GP [9] | -GP$_{\text{zero-cent}}$ [17] | -Dropout [19] | -SS [5] | $\overline{\Delta\text{PA}}$ |
|---|---|---|---|---|---|---|---|---|
| SN DCGAN | ✗ | 7.6 | 7.5 | 7.5 | 7.5 | 7.9 | — | 0.2 |
| NS Loss | feat | 7.8 | 7.7 | 7.7 | 7.7 | **7.9** | — | |
| SA GAN (sBN) | ✗ | 8.4 | — | 8.5 | 8.5 | 8.7 | 8.6 | 0.1 |
| Hinge Loss | feat | 8.6 | — | 8.6 | 8.7 | 8.7 | **8.8** | |
| | $\overline{\Delta\text{PA}}$ | 0.2 | 0.2 | 0.2 | 0.2 | 0.0 | 0.2 | |

## S8 Network Architectures and Hyperparameter Settings

In this work we exploit the implementation provided by [15, 13][3] and [22][4]. For the experiments, we run on single GPU (Nvidia Titan X).

### S8.1 Network Architectures

**SN DCGAN.** Following [16] for spectral normalization (SN), we adopt the same architecture as in [13] and present its configuration in Table S8. The input and feature (i.e., feat$_{\text{N}/2}$, feat$_{\text{N}/4}$ and feat$_{\text{N}/8}$) space augmentations respectively take place at the input of the layers with the index 0, 2, 4 and 6. In case of dropout, it is applied to the same intermediate layers plus the output of the layer 0. For Table 1 in the main paper, we pick the feat$_{\text{N}/8}$ for all evaluated datasets, whereas Sec. S4.1 presents an ablation study on the augmentation space.

**SA GAN (sBN).** The ResNet-based discriminator and generator architectures tailored for CIFAR10, CELEBA-HQ and T-ImageNet are presented in Table S9 and S11, respectively. Taking the ResNet architecture in [9] for CIFAR10, in [13] for CELEBA-HQ and [3] for IMAGENET as the baseline, we adapt them by adding the SN and self-attention as proposed in [22]. For the residual and non-local blocks we use the implementation provided by [22]. As we target unsupervised GAN, the conditional batch normalization (BN) used by the generator's residual blocks only takes the input noise vector $z$ as the conditioning, namely, self-modulation BN (sBN) [4].

For CIFAR10, we have considered the input and feature (i.e., feat$_{\text{N}/2}$ and feat$_{\text{N}/4}$) space augmentations which respectively take place at the input of the residual blocks with the index 0, 2 and 4, see Table S9-(a). Note that both residual blocks with the index 3 and 4 have their feature maps of dimension $N/4$. We experiment with the feature space augmentation on both of them. They differ little in performance, thereby we only report the result of the feature space augmentation at the residual block 4 in Table 1 of the main paper.

For CELEBA-HQ, we empirically observe that it is beneficial to start from a convolutional layer rather than a residual block at the discriminator. Apart from input and feat$_{\text{N}/8}$ space augmentation reported in Table 1 of the main paper, we have also experimented the other feature space augmentations that take place at the input of each residual block, see Table S10. At the spatial dimension $N$, we only

**Table S8:** SN DCGAN.

**(a)** Discriminator

| # | Configuration per Layer |
|---|---|
| 0 | $3 \times 3$ stride 1 SN Conv, $\mathtt{ch} = 64$, lReLu |
| 1 | $4 \times 4$ stride 2 SN Conv, $\mathtt{ch} = 128$, lReLu |
| 2 | $3 \times 3$ stride 1 SN Conv, $\mathtt{ch} = 128$, lReLu |
| 3 | $4 \times 4$ stride 2 SN Conv, $\mathtt{ch} = 256$, lReLu |
| 4 | $3 \times 3$ stride 1 SN Conv, $\mathtt{ch} = 256$, lReLu |
| 5 | $4 \times 4$ stride 2 SN Conv, $\mathtt{ch} = 512$, lReLu |
| 6 | $3 \times 3$ stride 1 SN Conv, $\mathtt{ch} = 512$, lReLu |
| 7 | SN Linear 1 output |

**(b)** Generator

| Configuration per Layer |
|---|
| Linear $h/8 \times w/8 \times 512$ output, BN, ReLU |
| $4 \times 4$ stride 2 DeConv, $\mathtt{ch} = 256$, BN, ReLU |
| $4 \times 4$ stride 2 DeConv, $\mathtt{ch} = 128$, BN, ReLU |
| $4 \times 4$ stride 2 DeConv, $\mathtt{ch} = 64$, BN, ReLU |
| $3 \times 3$ stride 1 Deconv, $\mathtt{ch} = 3$, Tanh |

**Table S9:** SA GAN for CIFAR10.

**(a)** Discriminator

| # | Configuration per Layer |
|---|---|
| 0 | ResBlock, down, $\mathtt{ch} = 128$ |
| 1 | Non-Local Block $(16 \times 16)$ |
| 2 | ResBlock, down, $\mathtt{ch} = 128$ |
| 3 | ResBlock, $\mathtt{ch} = 128$ |
| 4 | ResBlock, $\mathtt{ch} = 128$ |
| 5 | ReLU, Global sum pooling |
| 6 | SN Linear 1 output |

**(b)** Generator

| Configuration per Layer |
|---|
| SN Linear $4 \times 4 \times 128$ output |
| ResBlock, up, $\mathtt{ch} = 128$ |
| ResBlock, up, $\mathtt{ch} = 128$ |
| Non-local Block $(16 \times 16)$ |
| ResBlock, up, $\mathtt{ch} = 128$ |
| BN, RELU |
| $3 \times 3$ stride 1 SN Conv. $\mathtt{ch} = 3$, Tanh |

**Table S10:** SA GAN for CELEBA-HQ.

**(a)** Discriminator

| # | Configuration per Layer |
|---|---|
| 0 | $3 \times 3$ stride 1 SN Conv, $\mathtt{ch} = 64$ |
| 1 | ResBlock, down, $\mathtt{ch} = 128$ |
| 2 | ResBlock, down, $\mathtt{ch} = 128$ |
| 3 | Non-Local Block $(32 \times 32)$ |
| 4 | ResBlock, down, $\mathtt{ch} = 256$ |
| 5 | ResBlock, down, $\mathtt{ch} = 256$ |
| 6 | ResBlock, down, $\mathtt{ch} = 512$ |
| 8 | ReLU, Global sum pooling |
| 9 | SN Linear 1 output |

**(b)** Generator

| Configuration per Layer |
|---|
| SN Linear $4 \times 4 \times 512$ output |
| ResBlock, up, $\mathtt{ch} = 512$ |
| ResBlock, up, $\mathtt{ch} = 256$ |
| ResBlock, up, $\mathtt{ch} = 256$ |
| Non-local Block $(32 \times 32)$ |
| ResBlock, up, $\mathtt{ch} = 128$ |
| ResBlock, up, $\mathtt{ch} = 64$ |
| BN, RELU |
| $3 \times 3$ stride 1 SN Conv. $\mathtt{ch} = 3$, Tanh |

report the result of input space augmentation, whereas the feature space augmentation at the first residual block delivers a similar performance. Augmenting the input of the last residual block benefits from the first warm-up mechanism presented in Sec. S3.3, otherwise the discriminator can fail after augmentation progression.

For T-ImageNet, we have experimented with the augmentation space at both the input and $\mathtt{feat}_{16}$ (at the input of the 3rd residual block) and reported their performance in Table 1 of the main paper. It is beneficial to use the second warm-up mechanism introduced in Sec. S3.3. Comparing with the other datasets, the synthesis quality on T-ImageNet is still poor. Single GPU simulation with $64$ samples per batch is not enough in this case. Large-scale simulation as in [3], though demanding a large amount of resources, would be of interest.

## S8.2 Network Training Details

The training details across the datasets (i.e., F-MNIST, CIFAR10, CELEBA-HQ and T-ImageNet) and architectures (i.e., SN DCGAN, and SA GAN) are summarized in Table S12. For both architectures, the decay rate of the (s)BNs at the generator is set to $0.9$. During the evaluation phase, the generator

**Table S11:** SA GAN for Tiny-IMAGENET.

**(a)** Discriminator

| # | Configuration per Layer |
|---|---|
| 1 | ResBlock, down, $\mathtt{ch} = 64$ |
| 2 | Non-Local Block ($32 \times 32$) |
| 3 | ResBlock, down, $\mathtt{ch} = 128$ |
| 4 | ResBlock, down, $\mathtt{ch} = 256$ |
| 5 | ResBlock, down, $\mathtt{ch} = 512$ |
| 6 | ResBlock, $\mathtt{ch} = 512$ |
| 8 | ReLU, Global sum pooling |
| 9 | SN Linear 1 output |

**(b)** Generator

| Configuration per Layer |
|---|
| SN Linear $4 \times 4 \times 512$ output |
| ResBlock, up, $\mathtt{ch} = 512$ |
| ResBlock, up, $\mathtt{ch} = 256$ |
| ResBlock, up, $\mathtt{ch} = 128$ |
| Non-local Block ($32 \times 32$) |
| ResBlock, up, $\mathtt{ch} = 64$ |
| BN, RELU |
| $3 \times 3$ stride 1 SN Conv. $\mathtt{ch} = 3$, Tanh |

uses the moving averaged mean and variance to produce synthetic samples, thereby being independent of batch size.

### S8.3   Other Hyperparameter Settings

**Comparison with SotA on Human Face Synthesis.**   For CELEBA ($64 \times 64$), we used the same network architecture as T-ImageNet. This network is not as tailored as PG-GAN [11] and COCO-GAN [14] for human face synthesis. Unlike the other experiments, we followed the FID evaluation of COCO-GAN [14] for the sake of fair comparison. The augmentation space is at $\mathtt{feat_8}$ (the input of the 4th residual block). The hyperparameter setting for the $D$ and $G$ optimizers is: $\mathtt{lr_d} = 0.0004$, $\mathtt{lr_g} = 0.0001$, $\beta_1 = 0$, $\beta_2 = 0.999$, $\mathtt{iter_d}/\mathtt{iter_g} = 1$ and 1m training iterations.

**Regularization Techniques in Table 3**   In Sec. 4 of the main paper, we have experimented with a diverse set of regularization techniques and reported the FIDs in Table 3. Their settings are as follows:

For $\mathtt{Label}$ $\mathtt{smooth.}$, we followed the one-side label smoothing presented in [18] smoothing the positive labels from 1 to 0.9 and leaving the negative ones to 0 in the binary classification task of the discriminator.

The $\mathtt{GP}$ from [9] and the zero-centered alternative $\mathtt{GP_{zero\text{-}cent}}$ from [17] are implemented by exploiting the publicly available code in `https://github.com/igul222/improved_wgan_training` and `https://github.com/rothk/Stabilizing_GANs`. The weighting parameter for $\mathtt{GP}$ and $\mathtt{GP_{zero\text{-}cent}}$ is respectively set to 1 and 0.1 as suggested by [13, 17].

When combining $\mathtt{GP}$ with $\mathtt{PA}$, we adjust its weighting factor whenever kicking off a new augmentation level, namely, gradually increasing the weighting factor from zero to its original value within 5k iterations. This is mainly because the new bit can flip the reference label. Such relaxation on the 1-Lipschitz constraint allows the discriminator to timely cope with the new augmentation bit. Using $\beta_2 = 0.99$ instead of $\beta_2 = 0.9$ stabilizes the training on SA GAN.

For $\mathtt{Dropout}$, we experimented with different keep rates and applied layers. From Table S2, we selected the best performing setting of the $\mathtt{Dropout}$ with the keep rate 0.7 applied on the feature space with the spatial dimension $N/4$.

For $\mathtt{SS}$, we used the same mini-batch construction as in [5] for computing the auxiliary rotation loss. The rotation loss is respectively added to the $D$ and $G$ loss with the weighting factors equal to 1.0 and 0.2 as suggested by [5]. The augmentation bits does not affect the reference label when constructing the rotation loss.

**WGAN-GP**   In Sec. S3.4, we additionally trained CIFAR10 on SN DCGAN with WGAN-GP. The learning rates $\mathtt{lr_d}$ and $\mathtt{lr_g}$ remain the same as that of NS loss, i.e., $2 \times 10^{-4}$, but with two discriminator steps per generator step. The two momentum parameters for the Adam optimizer change to $\beta_1 = 0$ and $\beta_2 = 0.9$. The GP is weighted by one.

**Table S12:** Training details for the experiments in this work.

| Hyper-parameters | SN DCGAN NS Loss | | | SA GAN (sBN) Hinge Loss | | |
|---|---|---|---|---|---|---|
| | F-MNIST | CIFAR10 | CELEBA-HQ | CIFAR10 | CELEBA-HQ | T-IMAGENET |
| $\beta_1$ | 0.5 | 0.5 | 0.5 | 0.0 | 0.0 | 0.0 |
| $\beta_2$ | 0.999 | 0.999 | 0.999 | 0.9 | 0.9 | 0.9 |
| $lr_d$ | $10^{-4}$ | $2 \times 10^{-4}$ | $2 \times 10^{-4}$ | $3 \times 10^{-4}$ | $3 \times 10^{-4}$ | $3 \times 10^{-4}$ |
| $lr_g$ | $4 \times 10^{-4}$ | $2 \times 10^{-4}$ | $2 \times 10^{-4}$ | $10^{-4}$ | $10^{-4}$ | $10^{-4}$ |
| $iter_d/iter_g$ | 1 | 1 | 1 | 1 | 1 | 1 |

## Footnotes

[1]Both $\mathbb{P}_d$ and $\mathbb{P}_g$ are absolutely continuous with respect to $\mathbb{P}_m$. Therefore, their densities exist.

[2]`https://github.com/boschresearch/PA-GAN/blob/master/pagan_ops.py`

[3]https://github.com/google/compare_gan

[4]https://github.com/brain-research/self-attention-gan