[Reviews · NeurIPS 2019]

Reviewer 1



The proposed approach of making the discriminator's task progressively harder to regularize GAN training appears novel, and makes sense. I have a relatively large reservation regarding datasets; all use small images by today's standards, CELEBA-HQ128 being the only one at its resolution and the rest in smaller resolutions. Given the success of progressive growing (Karras18) in pushing to high resolutions, I feel we're not seeing the entire picture here. Direct comparison to progressive GAN – and perhaps a combination of the two – would also be interesting. I realize large resolutions mean higher computational demands, but the comparison could be made also in smaller resolutions. Improvements over dropout appear modest, but to the authors' credit they clearly state they searched for and used the best-performing variant. It would be interesting to know how variable the results are based on which layer dropout is applied to. Overall, I think this is a solid contribution, but one that does not appear to bring dramatic new capability or understanding.

Reviewer 2



The paper does a nice job at reviewing some of the existing work for improving image quality and stabilizing the training. The PA method is novel and surprisingly effective, with performance demonstrated over a wide range of datasets and is complementary to existing methods. The theoretical investigation of PA is a welcome addition. I am particularly pleased that the authors developed the means for automatic (metric based) scheduling of the difficulty progression. This is the kind of detail that is often left to a hand tweaked schedule which makes reimplementation and adaption much more difficult, the work spent here will definitely benefit future research. The paper is to the point and easy to read. I expect that the method may even have applications outside of those investigated, where the gap in difficulty between the discriminator and generator is larger.

Reviewer 3



This paper introduces a novel regularization method (e.g. progressive augmentation) of the original GANs to avoid the overshooting of discriminators and improve the stability of GAN training. Instead of weakening or regularizing the discriminator, the idea is to augment the data samples or features with random bits to increase the discrimination task difficulty. In this way, it could prevent the discriminator from being overconfident and maintain a healthy competition, which would enable the generator to be continuously optimized. The augmentation could be progressively levelled up during the training by evaluating the kernel inception distance between synthetic samples and training data samples. The proposed method has been demonstrated on different datasets and compared with other regularization techniques. The results show a performance improvement of the progressive augmentation (though there is no noticeable increase in visual quality). The paper also shows the flexibility of the proposed method. The progressive augmentation could be used with other regularizers and the combination could have good performance. The future work focuses on the implementation in semi-supervised learning, generative latent modelling and transfer learning. Overall, the content of this paper is complete and rich and has good technical quality. The results are well analysed and evaluated, and the claims of the paper are supported. The clarity is good but could be better. The author's response has been taken into account.

[Author Response · NeurIPS 2019]

We thank the reviewers for their positive feedback, indicating that our method is novel, theoretically justified and leads
to improved results (all); acknowledging the solid contribution of this work (R1), its usefulness for future research and
practitioners (R2) and high significance to the study of adversarial learning (R3). Next, we address individual concerns.

**R1** [Despite decreasing FIDs, it's hard to pinpoint a noticeable increase in visual quality]: A lower FID can be attributed
to not only increased visual quality but also the improved sample diversity. By looking at generated images in Fig. 3
and Fig. S4 in the sup. mat., we observed that PA mostly increases the variation of samples. This is expected as PA
being a regularizer doesn't modify the GAN architecture, as in PG-GAN (Karras'18) or BigGAN (Brock'19), to directly
improve the visual quality. For further evaluation, we measured the MS-SSIM of 10k synthetic CELEBA-HQ samples.
PA reduces it from 0.283 to 0.266, where the MS-SSIM of 10k real samples is 0.263 and PG-GAN reported 0.283.
Thus we believe that the quality of our generated samples is comparable to the ones of SoTA models, such as PG-GAN.

**R1** [A solid contribution, but does not appear to bring dramatic new capability]: We believe PA introduces a novel
mechanism to balance the two-player game in GAN training and partially alleviates the need of fine hyper-parameter
tuning (such as D and G learning rates in Tab. S4 in the sup. mat. or number of training iterations in Fig. 1 below),
which is, as known by the practitioners, often the key in achieving good quality and diversity of synthetic GAN samples.
To showcase this effect of PA, in Fig. 1 we present a toy example on MNIST. Note that PA effectively regularizes the
training and enables continuous learning, maintaining a good sample diversity (FID stays $\sim 2.5$ across iterations).
Without PA, the training rapidly becomes unstable, leading to mode dropping (digit 1 occurs more frequently).

**R1** [Reservation regarding datasets, comparison with PG-GAN]: For our experiments we selected "a wide range of
important datasets" (R2), which serve as default benchmarks in image synthesis literature. Despite their different
resolutions ($28 \sim 128$) and numbers of classes (1, 10 and 200), PA leads to consistent improvement over SoTA models.
For instance, with the SoTA SA GAN across CIFAR10 ($32^2$), T-Imagenet ($64^2$) and CELEBA-HQ ($128^2$), PA improves
FID by 2.7, 2.9 and 2.4, respectively. Thus, we believe the similar trend would occur for high resolution image
generation. Unfortunately, due to high computational load, we were not able to finish high resolution experiments on
time. At the smaller resolution 64 of CELEBA, PA improves the SA GAN FID of 4.11 to 3.35, being on a par with
COCO-GAN (Lin'19), FID of 3.57, that outperforms PG-GAN at the resolution 128, i.e., FID 5.74 vs. 7.30.

**R1** [How variable are the results based on which layer the dropout applied to]: We report these results in Tab. 1 and Tab.
S2, S3 in the supp. material. Adding PA is beneficial independent of the dropout settings (keep rate and applied layer),
it helps to reduce the FID sensitivity to the dropout hyperparameter choice.

**R2** [Motivation of adding Gaussian noise]: We agree that
the initial motivation of adding noise in [1, 29] was to
ensure a joint support of the data and model distributions,
which results in a harder task for discriminator. We will
clarify this point in the related work.

**Table 1:** PA with Dropout on CIFAR10 with SN DCGAN.

| Dropout | input(N) | | feat$_{N/2}$ | | feat$_{N/4}$ | | feat$_{N/8}$ | |
|---|---|---|---|---|---|---|---|---|
| PA(feat$_{N/8}$) | ✗ | ✓ | ✗ | ✓ | ✗ | ✓ | ✗ | ✓ |
| Keep rate 0.9 | 26.4 | 22.6 | 25.1 | 21.9 | 23.4 | 21.2 | 24.6 | 21.6 |
| 0.7 | 28.0 | 22.9 | 25.6 | 21.3 | _22.1_ | **20.6** | 24.4 | 22.5 |
| 0.5 | 27.1 | 23.1 | 25.9 | 22.3 | 23.1 | 21.2 | 24.0 | 22.1 |
| $\Delta$PA | 4.5 | | 3.7 | | 1.9 | | 2.3 | |

**R2** [Good to be more clear that the numbers reported
do not use any label conditioning]: We thank R2 for the
suggestion. Our trained models indeed do not use any label information, while in the case of SS-GAN [6] plus PA
combination achieving the FID score of 14.9 on CIFAR10, comparable to the supervised case with large scale BigGAN
[4] training. We will make this point clear in Sec.4.

**Table 2:** PA with different regularizations on CIFAR10.

| Method | PA | GAN | Lab. sm. | GP | GP$_{zero\text{-}cent}$ | Dropout | SS | $\Delta$PA |
|---|---|---|---|---|---|---|---|---|
| SN DCGAN | ✗ | 26.0 | 25.8 | 26.7 | 26.5 | 22.1 | — | - |
| | input | 22.2 | 23.1 | 21.8 | 22.3 | 21.9 | — | 3.0 |
| | feat | 22.6 | 22.3 | 22.7 | 23.0 | **20.6** | — | 3.0 |
| SA GAN | ✗ | 18.8 | — | 17.8 | 17.8 | 16.2 | 15.7 | - |
| | input | 16.1 | — | 15.8 | 16.1 | 15.5 | **14.7** | 1.3 |
| | feat | 16.3 | — | 16.1 | 15.9 | 15.6 | 14.9 | 1.5 |

**R3** [Input space augmentation with other regularization
techniques]: We provide the requested results in Tab. 2.
PA is consistently beneficial when combining with other
regularization techniques, independent of augmentation
space. Additional FID gain can come along with fine
selection of the augmentation space. We will add the numbers to the paper.

**R3** [Clarity of Fig.1,2]: We thank R3 for the detailed comments and will integrate the suggested changes into the paper.

**R3** [The claim that D performs a binary classification seems disputable]: We agree that, strictly speaking, $D(x)$ aims to
learn the probability of $x$ being real/fake rather than to perform classification. However, $D(x)$ can also be regarded as
the sigmoid response of classification with cross entropy loss. We will adjust our claim in Sec. 3.1 to avoid confusion.

**Figure 1:** PA enables continuous learning and prevents mode collapsing to a subset of classes (e.g., digit 1).

[Meta-Review · NeurIPS 2019]

The paper proposes progressive augmentation for GANs and shows that it leads to stable training and improves FID consistently. The author response addressed some of the initial concerns, and all the reviewers lean towards accepting the paper. It's nice to see that the proposed technique appears to be complementary to other regularization schemes, so it has the potential to be more widely useful for other machine learning problems (the authors themselves mention this as one of the future directions). I encourage the authors to incorporate reviewer feedback into the final version.